# Splicing accuracy varies across human introns, tissues, age and disease

S. García-Ruiz [1,2,3,4,5], D. Zhang[3], E. K. Gustavsson [1,3,5], G. Rocamora-Perez[3], M. Grant-Peters[1,2,3,5], A. Fairbrother-Browne[1,2,3,5], R. H. Reynolds [3], J. W. Brenton[1,2,3,5], A. L. Gil-Martínez[6], Z. Chen [3,6,7], D. C. Rio [5,8,9], J. A. Botia[10], S. Guelfi[6], L. Collado-Torres [11,12] & M. Ryten [1,2,3,4,5] ✉

Alternative splicing impacts most multi-exonic human genes. Inaccuracies during this process may have an important role in ageing and disease. Here, we investigate splicing accuracy using RNA-sequencing data from >14k control samples and 40 human body sites, focusing on split reads partially mapping to known transcripts in annotation. We show that splicing inaccuracies occur at different rates across introns and tissues and are affected by the abundance of core components of the spliceosome assembly and its regulators. We find that age is positively correlated with a global decline in splicing fidelity, mostly affecting genes implicated in neurodegenerative diseases. We find support for the latter by observing a genome-wide increase in splicing inaccuracies in samples affected with Alzheimer's disease as compared to neurologically normal individuals. In this work, we provide an in-depth characterisation of splicing accuracy, with implications for our understanding of the role of inaccuracies in ageing and neurodegenerative disorders.

RNA splicing is a post-transcriptional process in which introns are excised from messenger RNA (mRNA) precursors, and exons are joined together to form mature mRNAs. RNA splicing occurs within the nuclei of cells by base pairing between multiple small nuclear ribonucleoproteins forming the spliceosome and the sequences signalling the intron boundaries, termed splicing signals[1–3].

In humans, ~95% of multi-exon genes are alternatively spliced. Alternative splicing (AS) occurs when different combinations of exons are alternatively, rather than constitutively, spliced and included within the final mRNA, resulting in multiple RNA structures encoded by the same gene[4–6]. During AS, splice site choice is largely regulated by

cis-acting splicing regulatory elements (SREs)[7–10] that can enhance or silence the recognition of adjacent introns and exons. Different RNA-binding proteins (RBPs) are then responsible for interacting with these SREs and so activate or repress intron splicing accordingly within specific cells and tissues.

AS is a complex process and, consequently, accurate recognition and excision of introns and alternative exons relies on overcoming multiple challenges. First, the spliceosome must identify the splicing signals, namely the 5' splicing signal (5'ss), the branch point sequence and the 3' splicing signal (3'ss). Together, these sequences approximately encompass 25 base pairs (bp) distributed across the intron. This

[1]UK Dementia Research Institute, University of Cambridge, Cambridge, United Kingdom. [2]Department of Clinical Neurosciences, School of Clinical Medicine, University of Cambridge, Cambridge, United Kingdom. [3]Department of Genetics and Genomic Medicine Research & Teaching, UCL GOS Institute of Child Health, London, United Kingdom. [4]NIHR Great Ormond Street Hospital Biomedical Research Centre, University College London, London, United Kingdom. [5]Aligning Science Across Parkinson's (ASAP) Collaborative Research Network, Chevy Chase, MD 20815, USA. [6]Department of Clinical and Movement Neuroscience, Queen Square Institute of Neurology, UCL, London, United Kingdom. [7]The Francis Crick Institute, 1 Midland Road, London NW1 1AT, United Kingdom. [8]Department of Molecular and Cell Biology, University of California, Berkeley, CA 94720, USA. [9]California Institute for Quantitative Biosciences, University of California, Berkeley, CA 94720, USA. [10]Departamento de Ingeniería de la Información y las Comunicaciones, Universidad de Murcia, Murcia, Spain. [11]Lieber Institute for Brain Development, Baltimore, MD 21205, USA. [12]Department of Biostatistics, Johns Hopkins Bloomberg School of Public Health, Baltimore, MD 21205, USA. ✉e-mail: mr2022@medschl.cam.ac.uk

sets a relatively large mutational target in which germline and somatic variants could appear, compromising the correct identification of exon-intron boundaries[11–14]. Genetic variation can also alter the SREs, which can jeopardise the correct binding of splicing-related RBPs to these sequences and, therefore, accurate splicing. Third, some intronic sequences can be long (reaching lengths above 1 million bp[15] in humans), increasing the risk of cryptic splicing sequences[3] that can serve as decoy splice sites for spliceosome selection. Lastly, as observable in all biological systems, this process is subject to stochastic variation[16–20].

Ensuring splicing accuracy, namely the fidelity with which the splicing machinery performs intron excision and exon ligation to form mature mRNAs, is crucial for producing functional proteins and maintaining cell homoeostasis[21–27]. While mechanisms such as the nonsense-mediated decay (NMD) can mitigate the impact of spurious mRNA transcripts[28–33], differential use of splice sites escaping this mechanism has demonstrated widespread dysregulation in a range of diseases[34], including Alzheimer's disease (AD)[35,36], and ageing[37].

Different studies have demonstrated a decline in age-related splicing accuracy in species such as *Mus musculus*[38], *Drosophila*[39], *C.Elegans*[40] and *Homo Sapiens*[41–43]. However, to the best of our knowledge, no study to date has evaluated the genome-wide accuracy of splicing from an intron-level perspective across multiple tissues and human samples (>14k), in the context of age, neurodegeneration and with expression changes of important RBPs and NMD factors. To address these questions, we used RNA-sequencing data provided by the Genotype-Tissue Expression v8[44] project, and studied and characterised splicing accuracy across >300k annotated introns and >3m novel splicing events. We found robust patterns in the distribution of splicing noise, reflecting the molecular architecture of spliceosome assembly and action. By combining RNA-sequencing data from RBP knockdown experiments[45] and CLIP-seq experiments[46], we investigated the role of RBP and NMD expression in tuning splicing noise and changing its distribution. Given that RBP expression levels are known to change with age in humans[47] and that NMD activity has been shown to decrease during ageing in other organisms[48], we studied the effect of age on splicing accuracy. We demonstrated that age is positively correlated with a decline in splicing fidelity and that, in the human cortex, it affects genes implicated in neurodegenerative diseases. Using publicly-available RNA-sequencing data from the fusiform gyrus of AD and neurologically normal individuals[49], we observed a significant increase in inaccurate splicing in the AD brain, affecting genes implicated in neurodegenerative diseases and synaptic functions. Finally, we evaluated the relative contribution of important RBPs and NMD factors to the presence of inaccurately spliced transcripts with increasing age and in AD. We found that a decrease in the expression levels of RBPs, implicated in post-transcriptional functions, as well as core components of the NMD machinery, contribute to an increase in inaccurate splicing with increasing age and in AD. Altogether, these results demonstrate that inaccurate splicing is detectable across human tissues and modelling its characteristics provides novel insights into age-related and neurodegenerative diseases in humans (Fig. 1).

## Results

### Novel donor and acceptor junctions are commonly detected and exceed the number of unique annotated introns by an average of 11-fold

Splicing events can be accurately detected from short-read RNA-sequencing data using split reads. Split reads are reads that map to the genome with a gapped alignment, indicating the excision of an intron. We focused on three classes of split reads: i) annotated exon-exon junction reads, which precisely match an intron within annotation (Ensembl v105), ii) novel donor junctions, where only

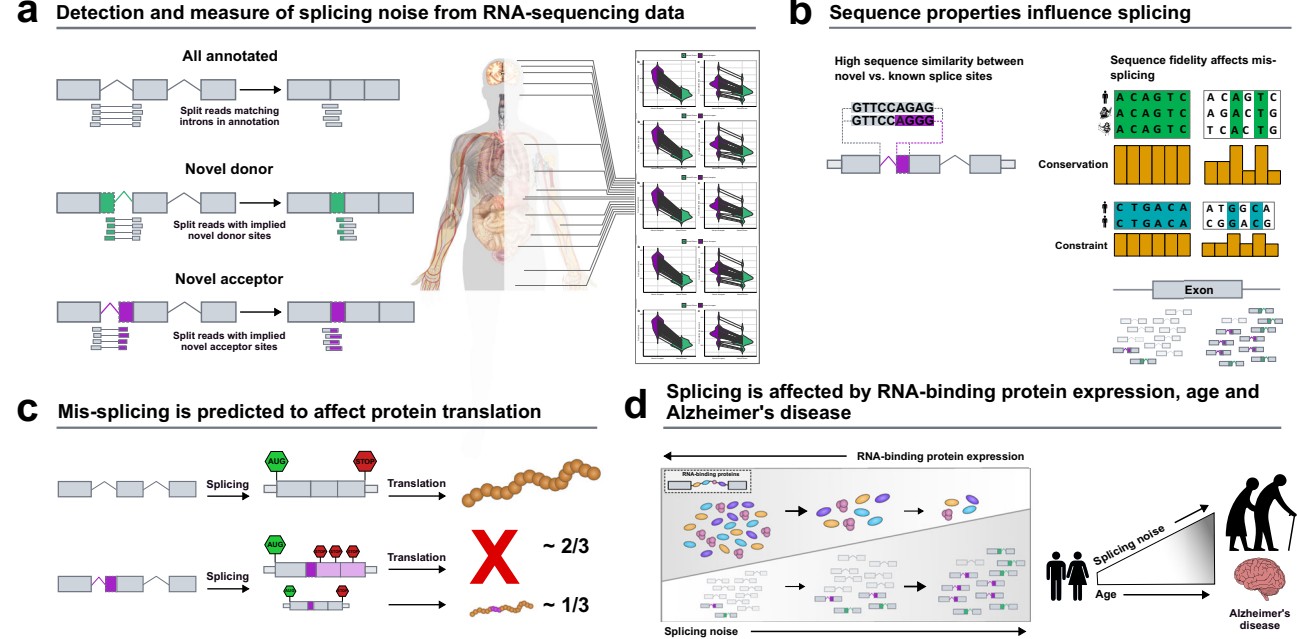

**Fig. 1 | Overview of the analyses performed in this study. a** We studied splicing accuracy through three classes of split reads spanning exon-exon junctions: annotated, novel donor and novel acceptor split reads. The RNA-sequencing dataset used originated from the Genotype-Tissue Expression (GTEx) project v8. In all 40 GTEx tissues studied, junctions from the novel acceptor category exceeded the number of unique novel donor junctions. **b** Novel splice sites from the novel donor and novel acceptor categories present high sequence similarity to annotated splice sites. High sequence fidelity in the vicinity of exon-intron junctions is required to accomplish accurate splicing. **c** Novel junctions associated with protein-coding transcripts are predicted to be deleterious in 2/3 of cases. **d** Reduced expression levels of the RNA-binding proteins responsible for sequence recognition appear to change splice site selection, which reduces the overall accuracy of the splicing process. Age is positively correlated with increases in splicing inaccuracies across multiple human tissues. Splicing inaccuracies are significantly higher in autopsy-confirmed Alzheimer's cases as compared to neurologically normal age-matched controls.

the implied 3'ss, namely acceptor site, matches an intron-exon boundary within annotation, and iii) novel acceptor junctions, where only the implied 5'ss, namely donor site, matches an exon-intron boundary within annotation (Fig. 1). We use the term "splicing accuracy" to refer to the study of splicing fidelity, primarily represented by unregulated errors that occur at low frequency in a global manner. To study splicing accuracy through the aforementioned three junction classes, we leveraged RNA-sequencing data processed by the relational database, IntroVerse[50], and originating from the Genotype-Tissue Expression (GTEx) Consortium[44] v8 data set. After quality-control processes, we used a subset of the data provided by IntroVerse relating to 324,956 annotated introns and 3,865,268 novel junctions (Supplementary Fig. 1). Briefly, this involved the discard of all split reads i) shorter than 25 bp, ii) located within unplaced sequences on the reference chromosomes, iii) overlapping with any of the regions included in the hg38 ENCODE[51] Blacklist, and/or iv) originating from introns targeted by the minor spliceosome[52].

We started by evaluating the extent to which each specific junction was shared between samples. We found that while the vast majority of novel junctions were unique to an individual or a very low number of individuals in all tissues (Supplementary Fig. 2), annotated introns were shared across a high number of samples (Supplementary Fig. 3). Next, we found that 268,988 (82.8%) annotated introns had at least one associated novel junction, with only 55,968 annotated introns appearing to be accurately spliced across all ~14k samples studied. Collectively, we detected 3,865,268 unique novel donor ($n = 1,582,593$) and acceptor junctions ($n = 2,282,675$), equating to 14 novel junctions per annotated intron. The detection of unique novel donor and acceptor junctions was a common finding across all tissues, with the highest numbers per sample found in Cell EBV-Transformed Lymphocytes tissue and the lowest in Whole Blood (Supplementary Fig. 4).

## Over 98% of novel donor and acceptor junctions are likely to be generated through inaccurate splicing

Unique novel junctions may represent novel transcripts[53], but given the high numbers detected, novel junctions could also be the product of splicing errors. To explore this, we leveraged the existence of multiple reference Ensembl transcriptome builds, namely v97 (May 2019) and v105 (June 2021), assuming an increased accuracy over their 2-year gap. For each tissue, we re-processed and re-annotated each split read provided by GTEx to the v97 and v105 annotation builds. We found that across all tissues, on average only 0.008 [0.005,0.012] of junctions defined as novel donor or acceptor junctions using v97 were reclassified as annotated introns in v105, and thus part of a transcript structure (Fig. 2a). Interestingly, we noted that the highest reclassification rates were observed amongst human brain tissues, on average 0.009 [0.008,0.012]. Given the widespread isoform diversity and alternative splicing found in frontal cortex[54], we extended our analysis in this body site and included Ensembl versions published from 2014 to 2021. The reclassification rate of novel junctions in frontal cortex decreased incrementally from 0.023 to 0.003 (Supplementary Fig. 5), consistent with previous studies reporting that the number of novel junctions entering annotation has been plateauing since 2013[55]. These findings suggest that the vast majority of novel junctions are generated through splicing inaccuracies, with on average <0.009 (< 0.9%) being explained by junctions originating from stable transcripts.

## Splicing inaccuracies are more common at acceptor than donor splice sites

The recognition of the donor splice site (5'ss) and acceptor splice site (3'ss) of an intron is performed by separate components of the splicing machinery[34,56,57]. We aimed to test whether splicing error rates at these splice sites also differed. To assess this, we compared the numbers of unique novel donor and acceptor junctions detected in each tissue to the numbers of unique annotated introns. We found that novel donor and acceptor junctions consistently accounted for the majority of unique junctions detected (70.8% [range: 58.2-79.1%]) and that the novel acceptor category exceeded the novel donor across the samples of all tissues (Fig. 2b). While we detected an average of 241,118 unique annotated introns across body sites, unique novel donor and acceptor junctions averaged 251,031 and 363,076, respectively. The relative sizes of junction categories robustly remained even after increasing the minimum number of supporting split reads required for a junction to be considered (Supplementary Fig. 6a), and after increasing the stringency in read alignment by raising the anchor length used (Supplementary Fig. 7a).

We reasoned that while splicing inaccuracies might generate high numbers of unique novel junctions in a given sample, each of these junctions would be expected to have a low number of associated reads. Consistent with this prediction, we found that novel donor and acceptor junctions together accounted for 0.32-1.08% of all junction reads whereas annotated introns accounted for 98.92-99.68% of the junction reads across all tissues evaluated (Fig. 2c, Supplementary Fig. 6b and Supplementary Fig. 7b). Focusing on frontal cortex, we found that annotated introns had a median read count of 2,695 supporting split reads, with novel donor and acceptor junctions having a median read count of only 2 split reads in both cases. These findings were replicated across all human tissues (Supplementary Table 1) and were consistent with novel junctions generated through splicing errors.

## High motif sequence similarity between novel splice sites and their annotated pairs explains inaccurate splicing

Sequences delineating intron boundaries are diverse and cryptic splice sites have the potential to induce splicing errors when present near them[58]. We applied the MaxEntScan[59] (MES) algorithm to assess the motif sequence similarity of all annotated and novel 5'ss and 3'ss to consensus representative sequences in humans. We found significant overlaps between the distribution of MES scores assigned to annotated versus novel splice sites, suggesting that the splicing machinery would be expected to recognise the latter (Supplementary Fig. 8).

Given that splice selection is likely to be a competitive process, we leveraged our paired data structure to compare MES scores between annotated introns and novel junction pairs (termed delta MES score). We found that the majority of novel 5'ss and 3'ss motif sequences were weaker than their paired annotated site, with 82.6% of novel 5'ss and 85.8% of novel 3'ss having positive delta MES scores (Fig. 3a, b, Supplementary Fig. 6c and Supplementary Fig. 7c). Moreover, novel 5'ss and 3'ss had a median delta value of 3.6 and 5.2, respectively, in keeping with the higher number of novel acceptor events as compared to novel donor junctions detected in all tissues, and similar MES scores to their annotated pairs. Overall, these results suggest that the strength of local splicing signals is not sufficient to guarantee accurate splicing[14,60].

## Novel junctions associated with protein-coding transcripts are predicted to be deleterious in 63.5% of cases

High sequence similarity between novel and annotated splice sites might be expected if these sites were located in close proximity. Thus, we analysed the relationship between annotated and novel splice sites focusing on the distribution of the latter within 30 bp upstream and downstream of annotated sites in frontal cortex tissue. We noted that: i) both novel 5'ss and 3'ss were located near paired annotated sites; ii) the distribution of splicing inaccuracies was different between annotated 5'ss (mode = −4bp/3 bp) and 3'ss (mode = −21bp/4 bp); and iii) splicing accuracy was highly asymmetric at annotated acceptor sites, with a very low error density upstream this intron-exon boundary, suggesting that this splicing pattern was driven by the AG exclusion zone[61,62]. These results were replicated across all tissues (Supplementary Table 2), consistent with novel junctions originating from splicing errors.

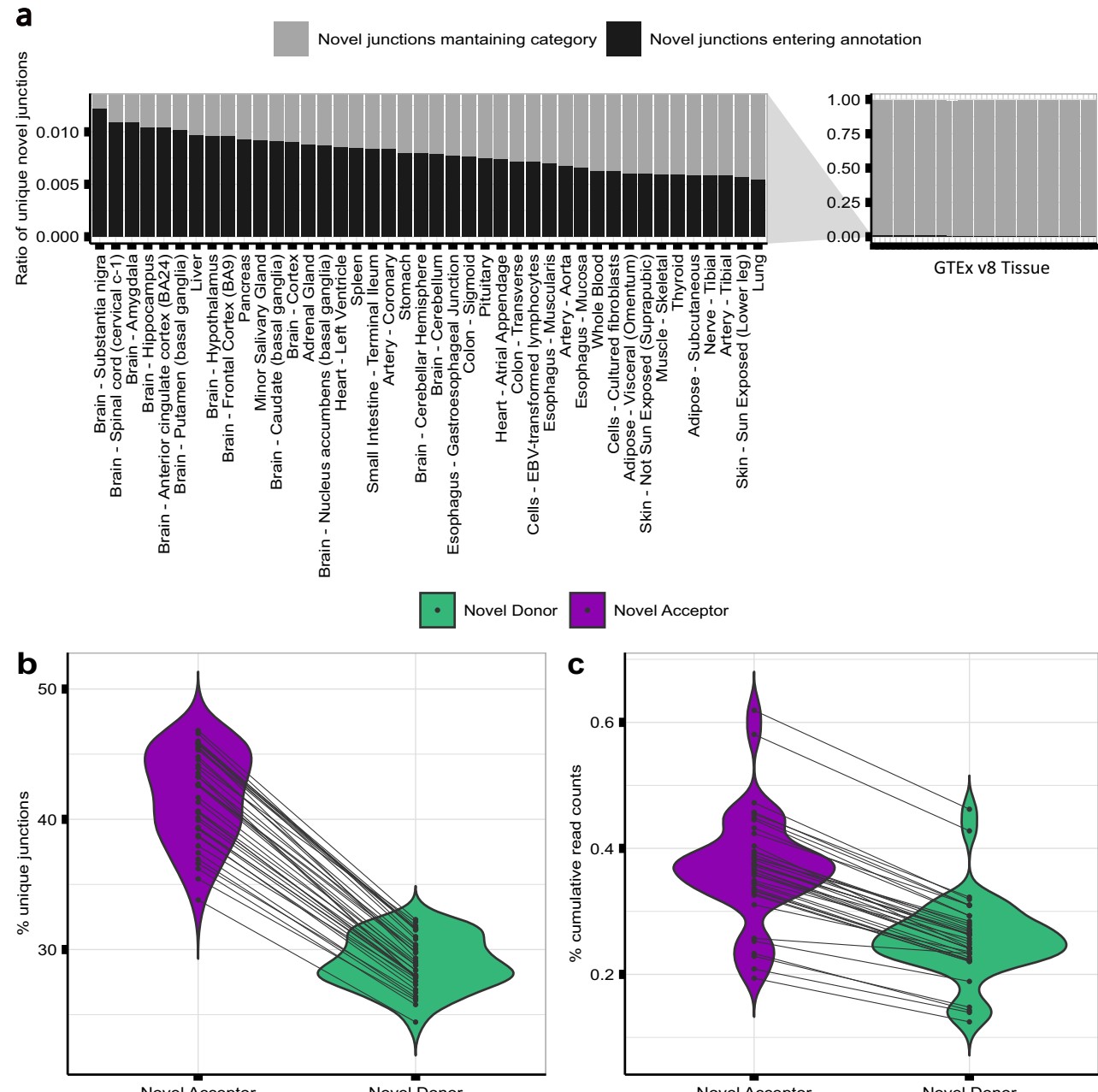

**Fig. 2 | Splicing accuracy can be measured using short-read RNA-sequencing data. a** Re-classification rate of novel split reads in Ensembl v97 compared to Ensembl v105 per GTEx tissue. Bars in black represent the ratio of split reads classified as novel junctions using Ensembl v97 that entered annotation as annotated introns in Ensembl v105. Bars in light grey represent the ratio of novel junctions in Ensembl v97 that maintained the novel annotation category in Ensembl v105. **b** Percentage of unique novel donor and novel acceptor junctions detected across the samples of each GTEx tissue (Ensembl v105). The crossing lines link the percentage of unique novel donor and novel acceptor junctions found within the same tissue. **c** Percentage of cumulative number of read counts that the novel donor and novel acceptor categories presented across the samples of each GTEx tissue (Ensembl v105). The crossing lines link the percentage of novel donor and novel acceptor split read counts detected within the same tissue.

We also observed regular splice site peaks occurring at 3 bp intervals, most apparent in novel acceptor events downstream of the paired annotated site, namely within annotated exons. Using data from frontal cortex tissue, we noticed that these peaks were only observed in novel events from protein-coding transcripts ($n = 20,605$) (Fig. 3c, d, Supplementary Fig. 6d and Supplementary Fig. 7d). To further explore this possibility, we studied the divisibility by 3, equating to the size of a codon, of the distances between each novel junction and their linked annotated 5′ss and 3′ss. Focusing on splice sites exclusively used in protein-coding transcripts in frontal cortex, this analysis demonstrated that 62.5% of all novel sites were located at distances not divisible by 3, implying that these

splicing events would result in deleterious frameshifts for downstream translation events. When focusing on each modulo3 value independently, we observed an overall preference to maintain the codon reading frame (mod3 = 0, 37.4%; mod3 = 1, 31.4%; mod3 = 2, 31.2%). Across all tissues, 63.55% of the novel junctions would likely disrupt the reading frame, supporting the view of novel junctions originating from splicing errors (Fig. 3e, Supplementary Fig. 6e and Supplementary Fig. 7e).

We hypothesised that the regular splice site peaks occurring at 3 bp intervals could be an evolved property of the genomic sequence, with cryptic splice sites preferentially located at these positions to prevent frame-shift events. Given that cryptic splice sites are known to have high

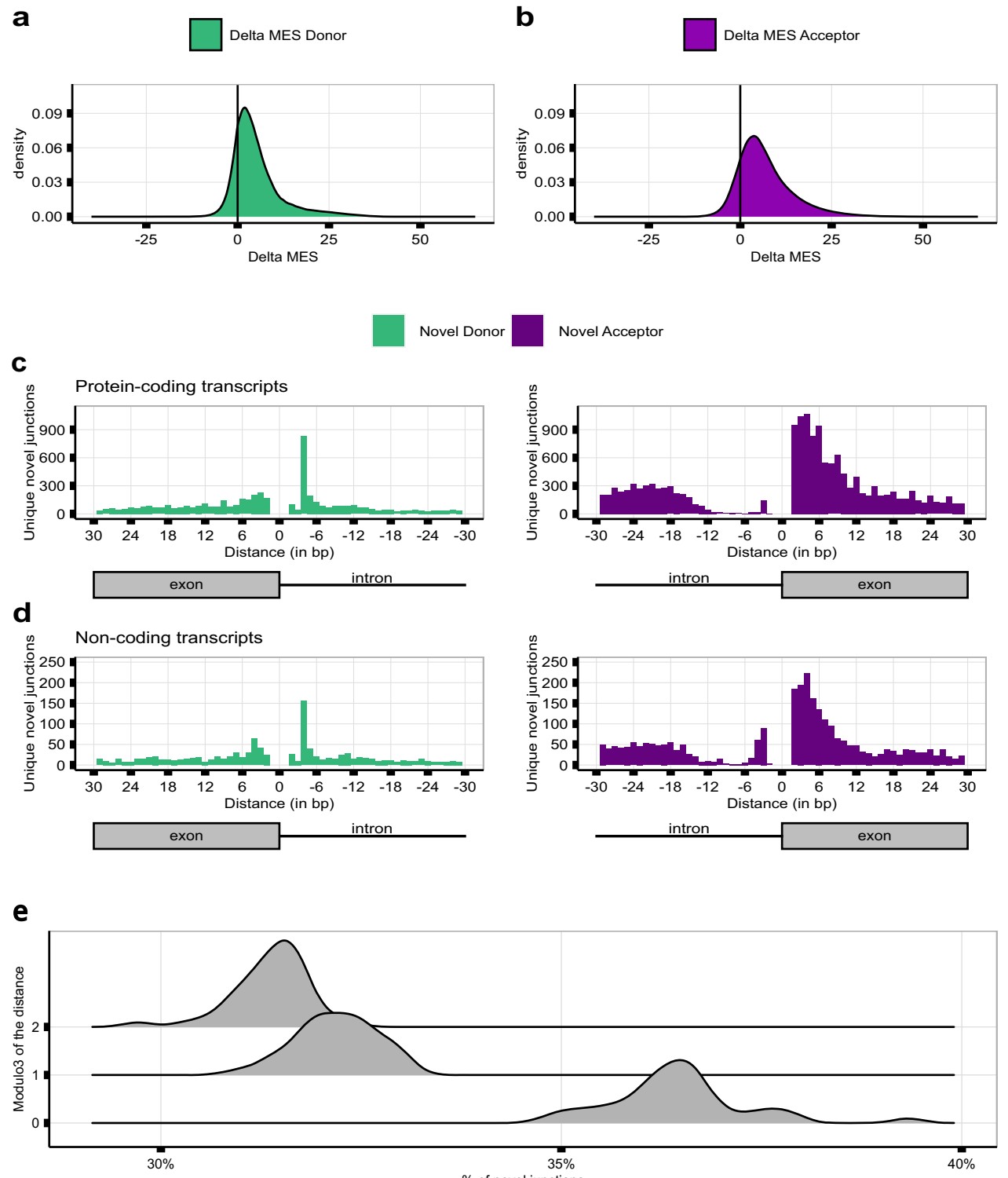

**Fig. 3 | Splicing inaccuracies can be explained by high sequence similarity between novel splice sites and their annotated pairs. a** MaxEntScan (MES) Delta scores between the 5'ss of the annotated introns and the 5'ss of their novel donor pairs across all tissues. **b** MES Delta scores between the 3'ss of the annotated introns and the 3'ss of their novel acceptor pairs across all tissues. **c, d** Distances lying between the novel splice site of each novel junction and its annotated intron pair in (**c**) protein-coding transcripts and (**d**) non-coding transcripts in frontal cortex tissue. **e** Modulo3 of the distances between each novel junction and its linked annotated intron to a maximum distance of 100 bp within MANE transcripts across all body sites.

motif sequence similarity to annotated splice sites[58], to test this hypothesis we obtained the delta MES of the novel splice sites located at distances divisible by three from their annotated pairs and compared them with those of the remaining novel junctions (namely those not located at distances divisible by three). We found no significant differences in motif sequence similarity across the novel junction types (one-tailed Wilcoxon Rank-sum test, P = 1, Supplementary Fig. 9). These findings suggested that the higher frequencies of novel acceptor

junctions at 3 bp intervals are not explained by genomic sequence properties, but are most likely to arise through a separate mechanism.

## Splicing accuracy varies across introns and is likely to be underestimated in bulk RNA-sequencing data

Next, we wondered if splicing fidelity varies across introns and genes across the genome. We used the Mis-Splicing Ratio measures to assess the frequency of splicing inaccuracies at both the 5′ss ($MSR_D$) and 3′ss ($MSR_A$) of each annotated intron. Focusing on frontal cortex brain tissue, we observed that while splicing errors were detected infrequently, with the $MSR_D$ and $MSR_A$ values highly skewed towards low values, there was considerable variation across introns ($MSR_D$ IQR = 5.7e-04; $MSR_A$ IQR = 1.6e-03). Furthermore, consistent with the overall higher detection of novel acceptors as compared to novel donor junctions, we observed a significant difference between the two $MSR_D$ and $MSR_A$ distributions (paired one-tailed Wilcoxon Rank-sum test, effect-size=0.09, P < 0.001) (Fig. 4a, Supplementary Fig. 6f, Supplementary Fig. 7f and Supplementary Table 3). Given that NMD activity would be expected to reduce the detection of splicing errors amongst mRNA transcripts, we compared MSR measures of annotated introns in protein-coding versus non-coding transcripts in samples from frontal cortex tissue after controlling for read depth (Supplementary Fig. 10). We found that splicing inaccuracies were more frequent amongst annotated introns from non-coding transcripts as compared to those from coding transcripts, at both their 5′ss (paired one-tailed Wilcoxon Rank-sum test, effect-size=0.17, P < 0.001) and 3′ss (paired one-tailed Wilcoxon Rank-sum test, effect-size=0.19, P < 0.001) (Fig. 4b, c, Supplementary Fig. 6g and Supplementary Fig. 7g), suggesting that the frequency of splicing errors is likely to be underestimated. These findings were validated across all tissues (Supplementary Table 3).

## High sequence fidelity in the vicinity of exon-intron junctions is required to maintain splicing accuracy

Given the variability found in splicing fidelity across introns, we wanted to identify features that could influence its generation. Focusing on frontal cortex tissue, we built two zero-inflated poisson regression models to predict the rate of splicing inaccuracies as defined by $MSR_D$ and $MSR_A$ values. We used as predictors different features of each annotated intron and the gene from which it originated. This analysis yielded three main findings. Firstly, we found that gene-level features had a small but significant effect on splicing accuracy. Increases in associated transcript number and protein-coding frequency predicted a reduction in splicing errors, suggesting that splicing inaccuracies within genes with high transcript diversity might be energetically costly for organisms[20,63] and so selected against (Fig. 4d). Secondly, this analysis provided support for splice site intercommunication[3,34,64], with sequence properties at both splice sites impacting splicing fidelity. Interestingly, we found that higher conservation scores (phastCons17) in genomic regions flanking the 5′ss and 3′ss were associated with lower splicing error rates (5′ss: $MSR_D$ = −0.9 [−0.8, −1.01]; $MSR_A$ = −0.85 [−0.76, −0.95]) (3′ss: $MSR_D$ = −0.78 [−0.69, −0.88]; $MSR_A$ = −0.78 [−0.7, −0.87]). Similarly, highly constrained sequences amongst humans (mean context-dependent tolerance, CDTS) in the vicinity of 5′ss and 3′ss, were associated with lower splicing error rates at both splice sites (5′ss: $MSR_D$ = 1.01 [1.01, 1.02]; $MSR_A$ = 1.01 [1.01, 1.02]) (3′ss: $MSR_D$ = 1.01 [1, 1.01]; $MSR_A$ = 1.01 [1.01,1.02]). Overall, these results suggested that low sequence variation within intronic sequences flanking exon-intron junctions are associated with increased splicing fidelity.

## Splicing accuracy is affected by RNA-binding protein expression changes

To better understand the factors influencing splicing accuracy across tissues, we expanded our analysis in frontal cortex tissue to all body sites. Interestingly, this identified unexpectedly high variation in the effect of sequence conservation on splicing accuracy (as captured by the beta coefficient) across tissues (Fig. 5a, b), despite the conservation scores being identical across body sites. We hypothesised that this finding could have arisen because we had not accounted for the impact of somatic variation, and that this could alter critical splicing sequences and cis-acting SREs, hence causing changes in their recognition by RBPs and resulting in splicing errors. To test for this possibility, we compared MSRs between sun-exposed and not-sun-exposed skin across common annotated introns on the basis that sun-exposed skin is known to have higher rates of somatic mutations[65]. However, we did not find any significant differences in MSRs from annotated introns between these two tissues (two-tailed Wilcoxon Rank-sum test $MSR_D$ P = 0.14; two-tailed Wilcoxon Rank-sum test $MSR_A$ P = 0.25, Supplementary Fig. 11). Based on these findings, we considered if the observed tissue-specific expression levels of RBPs involved in splicing processes across body sites (Supplementary Fig. 12), could explain the variable effect of sequence conservation observed on MSRs. To explore this possibility, we analysed ENCODE data involving knockdowns of 54 genes related to splicing regulation, spliceosome assembly, exon-junction complex (EJC) recognition[45] and NMD (Supplementary Fig. 13). This analysis yielded three main findings. Firstly, it revealed a significant increase in MSRs in samples with gene knockdowns compared to untreated controls for 90% ($MSR_D$ FDR < 0.001, n = 49) and 94% ($MSR_A$ FDR < 0.001, n = 51) of the 54 genes considered, respectively. Knockdowns of the splicing machinery and EJC components tended to have a greater effect on 3′ss than 5′ss (mean $MSR_D$ effect-size = 0.10 [0.02,0.39]; mean $MSR_A$ effect-size = 0.12 [0.01, 0.62]), except for 6 genes, including *SAFB2*, which is not thought to impact on splicing and so was used as a negative control (Supplementary Tables 4, 5). Notably, *AQR, EFTUD2, HNRNPC, MAGOH, SF3A3, SF3B4, U2AF1* and *U2AF2* knockdowns resulted in the highest increases in 5′ss and 3′ss MSRs (Fig. 5c). Secondly, knocking down components of the NMD pathway, such as *UPF1* and *UPF2*, produced a detectable but modest increase in the levels of splice-site noise as compared to the knockdown of other RBPs such as *MAGOH*, a core component of the EJC involved in the activation of the NMD pathway[66]. We also found that 3′ splicing errors that were only evident in the context of NMD knockdown, were generated from annotated introns which had significantly lower phastCons17 and MES values (phastCons17: one-tailed Wilcoxon Rank-sum test, effect-size=0.07, P < 0.001; MES: one-tailed Wilcoxon Rank-sum test, effect-size=0.03, P = 0.01) (Supplementary Fig. 14). Interestingly, across the *UPF1* and *UPF2* knockdown experiments, analysis of the genomic distances from novel junctions to their annotated pairs still demonstrated regular 3 bp peaks in protein-coding transcripts, indicating that NMD was not preferentially acting at positions that could generate frameshift events (mod3 = 1, mod3 = 2) (Supplementary Fig. 15 and Supplementary Fig. 16). Thirdly, this analysis revealed distinct patterns in MSRs distribution depending on the gene targeted. For instance, knocking down AQR expression led to a remarkably high number of splicing inaccuracies within 15-200 bp upstream of the annotated acceptor site (Fig. 6a and Supplementary Fig. 17), including weaker 3′ss selection (one-tailed Wilcoxon Rank-sum test, effect-size=0.14, P < 0.001) (Fig. 6b), suggesting that the spliceosome was no longer able to distinguish splicing signals at acceptor sites accurately. *U2AF2* knockdowns resulted in a relatively high number of splicing inaccuracies within 15-30 bp upstream of the acceptor sites (Fig. 6a), including the selection of weaker novel 3′ss (one-tailed Wilcoxon Rank-sum test, effect-size=0.02, P < 0.001) (Fig. 6b). We further investigated the role of RBPs in splicing errors by jointly analysing each knockdown experiment with corresponding CLIP-seq data[46] for 15 RBPs related to splicing regulation and spliceosome assembly. Importantly, we found that annotated introns with the highest levels of MSRs when a given RBP was knocked down, were also those introns with higher densities

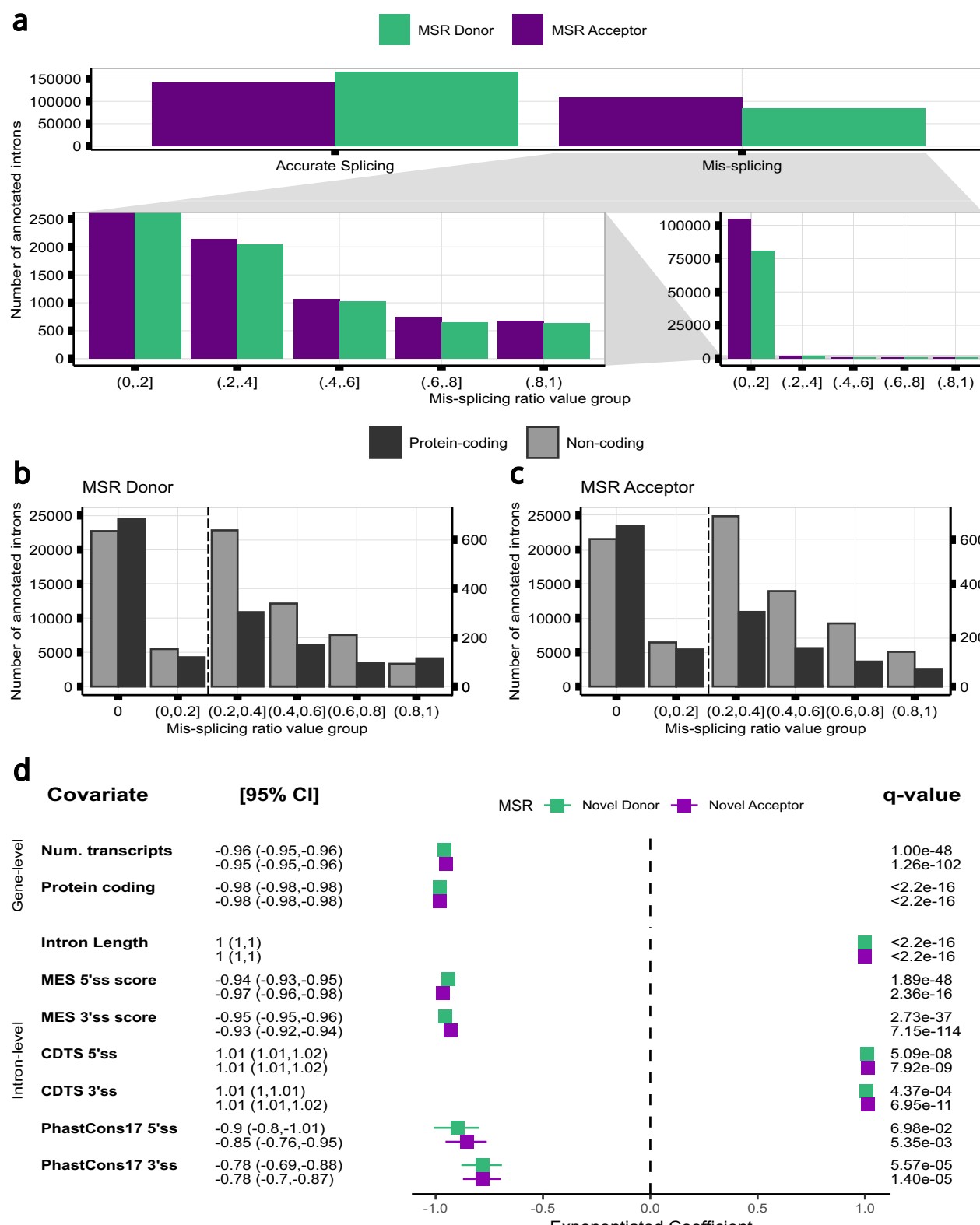

**Fig. 4 | Splicing inaccuracies vary across introns and are impacted by local sequence properties. a** Mis-splicing Rates (MSRs) at the 5′ and 3′ss of the annotated introns (*n* = 251,042) from frontal cortex samples (*n* = 186). Bottom right: MSRs from inaccurately spliced introns across binned values. Bottom left: a zoomed-in view of the bottom right panel. **b, c** MSRs at the (**b**) 5′ and (**c**) 3′ss of the annotated introns from protein-coding (*n* = 55,358) and non-coding (*n* = 55,358) transcripts in samples from frontal cortex tissue. The black dashed vertical line separates the bars displayed under the two y-axes. Right y-scale: a zoomed-in view

of the left y-axis. **d** Exponentiated beta coefficients from the count model of two zero-inflated poisson regression models (poisson family, log link function) to predict MSRs at the donor and acceptor splice sites, respectively, from the annotated introns (*n* = 224,189) in frontal cortex samples (*n* = 186). P-values from each ZIP model were corrected for multiple testing using the Benjamini-Hochberg method, resulting in q-values (error bars represent adjusted standard errors from each estimated coefficient; statistical tests were two-sided, with significance assessed at q < 0.05; *n* = 186 biologically independent replicates).

of RBP binding sites (Fig. 6c). This finding was significant at both donor and acceptor sites for all 15 RBPs analysed ($MSR_D$, Pearson's Chi-squared test, P < 0.001) ($MSR_A$, Pearson's Chi-squared test, P < 0.001) (Supplementary Table 6). Taken together, these findings indicate a direct link between reduced RBP expression and increased levels of splicing inaccuracies.

## Increasing age is associated with increasing levels of inaccurate splicing

Previous studies reported an overall reduction in the expression of multiple RBPs with age[47,67–72], producing associated changes in splicing accuracy[71]. We formally assessed this in the GTEx dataset, and found that the expression levels of 107 RBPs (FDR < 0.04) and 5 essential NMD genes[73] (FDR < 0.02) decreased with age in multiple tissues (Supplementary Fig. 18). Focusing on brain tissue alone, 40% of the 115 RBPs studied had decreased expression levels with age (FDR < 0.04) (Supplementary Fig. 18b). Given these findings, we investigated age-related increases in splicing inaccuracies. We grouped samples for each body site into 2 extreme age clusters, 20-39 and 60-79 years and, after controlling for potential confounding covariates, we selected a set of 139,419 annotated introns shared across age groups and body sites (Supplementary Fig. 19 and Supplementary Fig. 20). We found that $MSR_D$ values in the 60-79 age group were significantly higher than those in the 20-39 cluster in 12 of the 18 body sites analysed (effect size = 0.06 [0.006, 0.12]; FDR < 0.001). Similarly, $MSR_A$ values in the 60-79 age group were significantly higher than those in the 20-39 category in 13 of the 18 tissues assessed (effect size = 0.07 [0.02, 0.13]; FDR < 0.001). In both cases, the highest effect size was found in blood vessel tissue (Fig. 7a and Supplementary Table 7).

We also evaluated the relative contribution of individual NMD and RBP factors to the presence of inaccurately spliced transcripts with increasing age in blood, blood vessel and brain, selecting these tissues based on the high levels of age-related splicing effects (Fig. 7a). When we ranked all factors in terms of their contribution to age-related splicing fidelity, we noted that the top-ranked genes (top 10) for blood vessel and brain tissues were RBPs involved in splicing (Supplementary Fig. 21 and Supplementary Fig. 22), whereas for blood tissue it also included components of the EJC and the NMD pathway (Supplementary Fig. 23). This suggests that tissue-specific changes in the expression of RBP and NMD factors with age are likely to explain the increase in age-related splicing inaccuracies observed (Supplementary Tables 8-10).

Given the complexity of splicing in the human brain and the importance of age-related disorders affecting this organ, we further investigated the properties of introns with evidence of age-related increases in MSRs in brain. We identified 37,743 annotated introns of interest based on increasing $MSR_D$ or $MSR_A$ values with age. After assigning these introns to their unique genes (n = 12,408), we used Gene Ontology (GO) Enrichment analysis to determine if age-related increases in MSRs might have an impact on specific biological processes or pathways. Interestingly, this analysis identified significant enrichment in terms such as: neuron to neuron synapse (FDR < 0.001), tau protein binding (FDR = 0.006) and dendritic spine (FDR < 0.001) (Fig. 7b). Since the former term suggested that splicing inaccuracies might affect neurons more than other cell types, we assessed cell-type specific expression of RBPs in the human brain. Using single-nucleus RNA-sequencing data from the Allen Brain Atlas covering multiple cortical regions[74], we investigated the cell-type specificity of 111 splicing-regulator and spliceosomal RBPs[45] across all major cell types. We found that splicing-regulator RBPs were more highly expressed than would be expected by chance in oligodendrocyte precursor cells, 4 subtypes of GABAergic neuron and 5 subtypes of glutamatergic neuron (Fig. 7c, Supplementary Fig. 24 and Supplementary Tables 11, 12). The enrichment of splicing-regulator RBPs within specific neuronal cell types suggests that neurons may be particularly sensitive to changes in RBP expression, and by extension, particularly vulnerable to age-related increases in splicing inaccuracies.

## Splicing accuracy decreases in genes enriched for synaptic functions in Alzheimer's Disease

Given that ageing is a primary risk factor for multiple neurodegenerative diseases in humans, including Alzheimer's Disease (AD)[75], we studied splicing accuracy in post-mortem human brain amongst neurologically unaffected individuals and those with AD. Using short-read RNA-sequencing data originating from the fusiform gyrus of 48 individuals[49], and after controlling for potential confounding covariates, we analysed splicing across 193,487 annotated introns in AD cases and controls (Supplementary Fig. 25 and Supplementary Fig. 26). This analysis demonstrated a genome-wide increase in the number of unique novel 5′ and 3′ splicing events and associated novel reads in AD cases as compared to control samples (Fig. 8a, b). We studied the distances between the novel splice sites and their annotated pairs and observed a higher frequency of novel junctions located at positions not divisible by 3 bp in the AD-affected samples, indicating a higher likelihood of frame-shift events that would be expected to be deleterious (Fig. 8c). Analysis of MSR measures also demonstrated significantly higher levels of MSRs in AD samples at both donor and acceptor sites ($MSR_D$ effect-size=0.027, one-tailed paired Wilcoxon signed rank test, P < 0.001; $MSR_A$ effect-size=0.0375, one-tailed paired Wilcoxon signed rank test, P < 0.001). Moreover, we noted that genes containing introns with higher MSRs at donor or acceptor splice sites in AD as compared to control samples (n = 15,231) were enriched for synaptic functions (Fig. 8d, e). Finally, we evaluated the relative contributions of the NMD machinery and RBP factors to splicing inaccuracies in the disease state by mirroring the approach followed with ageing. The results of this analysis indicated that correcting for the expression of RBPs and NMD factors reduced the apparent impact of disease on splicing noise (Supplementary Fig. 27), with RBPs being top-ranked (Supplementary Table 13).

## Discussion

Here we have shown that inaccurate splicing is common across human tissues and occurs near annotated intron-exon boundaries distinctively and predictably. Using the MSR, our own measure to quantify splicing inaccuracies at both splice sites, we found that this is higher at acceptor sites than at donor sites, and in non-coding transcripts at both sites in all tissues. We discovered that splicing fidelity varies across introns and tissues, and is predictable based largely on local sequence properties. Reduced expression of spliceosome components and regulators is a significant contributing factor to the variability in MSRs, as evidenced by in vitro knockdowns of RBPs and supporting CLIP-seq data, and in vivo with ageing. In the ageing human brain, splicing inaccuracies affect genes involved in neuronal function and proteostasis, with implications for age-related neurodegenerative disorders. Considering the latter, we observed a genome-wide increase in MSRs in the AD brain, affecting genes involved in synaptic functions and suggesting the key importance of splicing integrity in maintaining cognitive function.

One of the most striking and robust findings in this study was the consistently higher accuracy of 5′ss as compared to 3′ss recognition. This is likely to reflect intrinsic weaknesses and molecular differences in these processes. Initial recognition of the 5′ss of an intron is carried out by the U1 snRNP complex of the spliceosome. Even though their base-pairing interactions are often imperfect, this process is thought to be highly efficient[57,76,77]. In contrast, recognition of the 3′ end of introns requires cooperative binding of three interacting proteins to three neighbouring sequence motifs. Besides, a given 3′ss can be associated with more than one functional branch point[78]. Our findings support this view and suggest that this complexity makes this process particularly sensitive to errors.

There are a range of ways in which splicing errors could arise at both splice sites. Most simply, they could originate from genomic sequence variation due to germline and somatic mutations or

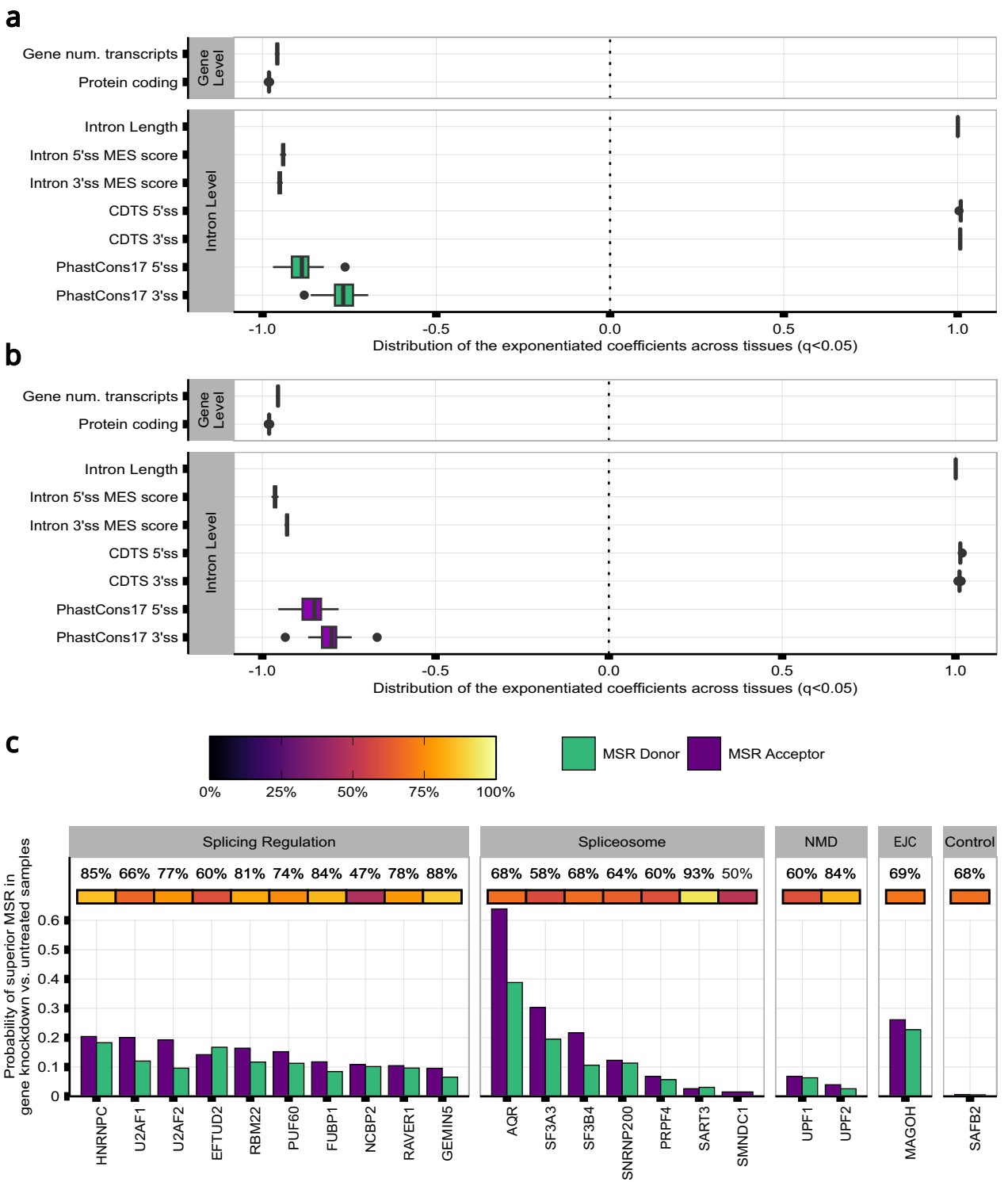

**Fig. 5 | Splicing inaccuracies vary across tissues and this could be explained by variable RNA-binding protein expression. a, b** Distribution of beta coefficient variation across the zero-inflated poisson regression (ZIP) models built to predict mis-splicing rates (MSRs) at the (**a**) donor (5'ss) and (**b**) acceptor (3'ss) splice sites of the annotated introns across the samples of each GTEx tissue (n = 40). P-values from the ZIP models were corrected for multiple testing using the Benjamini-Hochberg method, resulting in q-values. Only beta coefficient values for significant q values were considered for display. All statistical tests were two-sided, with significance assessed at q < 0.05. Box plots indicate median (middle line), 25th, 75th percentile (box) and 5th and 95th percentile (whiskers) as well as outliers (single points) of the distribution of the exponentiated beta coefficient values obtained across the n = 40 ZIP models built per MSR measure (one ZIP model per tissue and MSR measure, n = 80 ZIP models built in total). **c** Probability of superior MSRs at the 5'ss and 3'ss of the annotated introns in samples with the shRNA knockdown of each RBP as compared to untreated samples. The top heatmap track contains the knockdown efficiency of the associated protein.

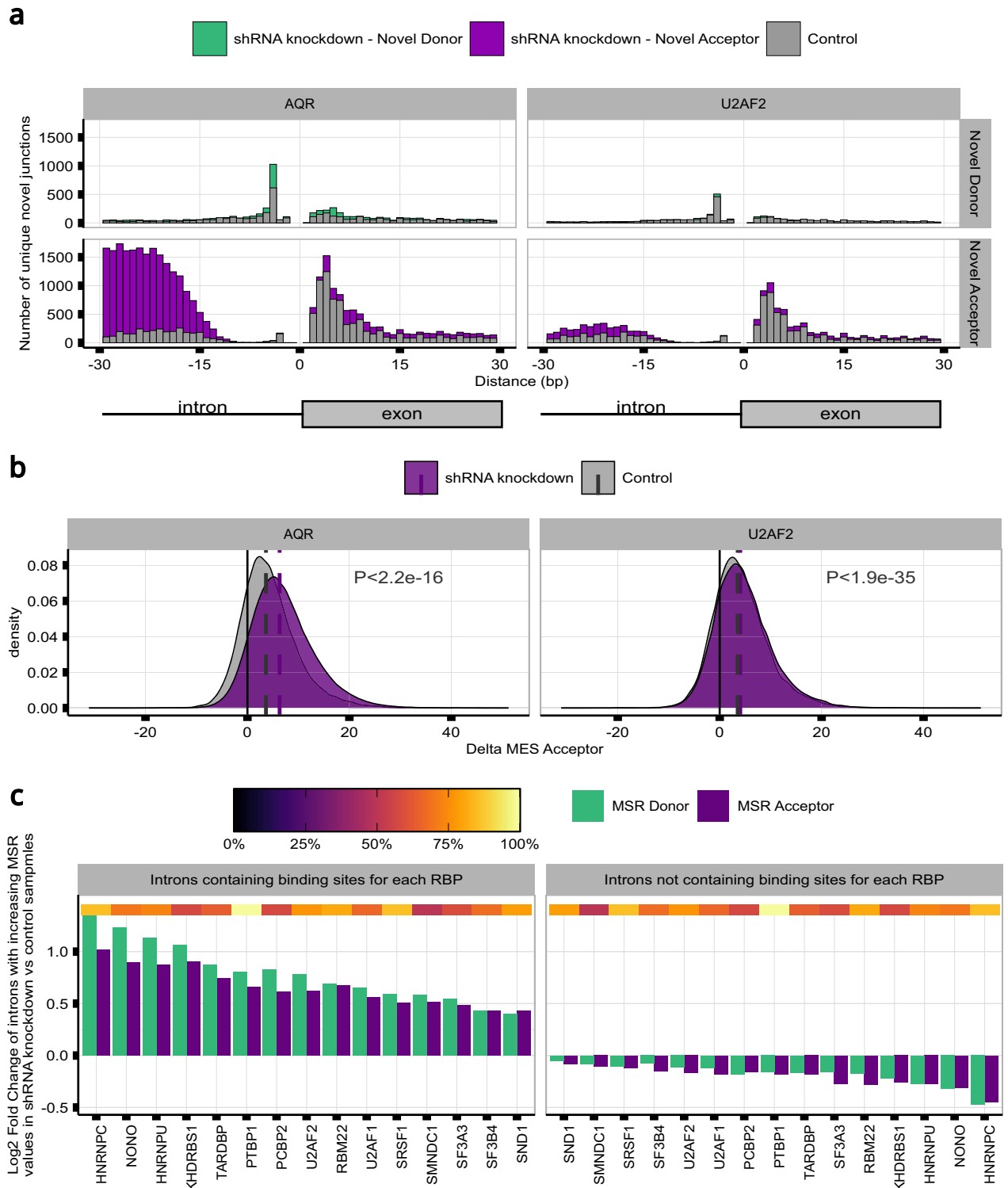

**Fig. 6 | shRNA knockdown of RNA-binding proteins (RBPs) produces different patterns of Mis-splicing ratios (MSRs) across introns, predominantly affecting annotated introns with higher RBP binding densities. a** Distances in base pairs from each novel donor and acceptor junction to their annotated intron pairs in shRNA knockdown experiments of *AQR* and *U2AF2*, respectively, as compared to samples from untreated controls. **b** MaxEntScan (MES) Delta scores between the novel 3'ss of each novel acceptor junction and its annotated intron pair in shRNA knockdown experiments of *AQR* and *U2AF2*, respectively, as compared to untreated controls. Dashed vertical lines represent the median value of each distribution and p-values are produced from a one-sided Wilcoxon Rank-sum test for differences between the two density distributions. **c** Log2 fold change in the MSRs of unique annotated introns following RBP knockdown and subclassified on the basis of their RBP binding densities as derived from CLIP-seq data. The top heatmap track contains the knockdown efficiency of the associated protein.

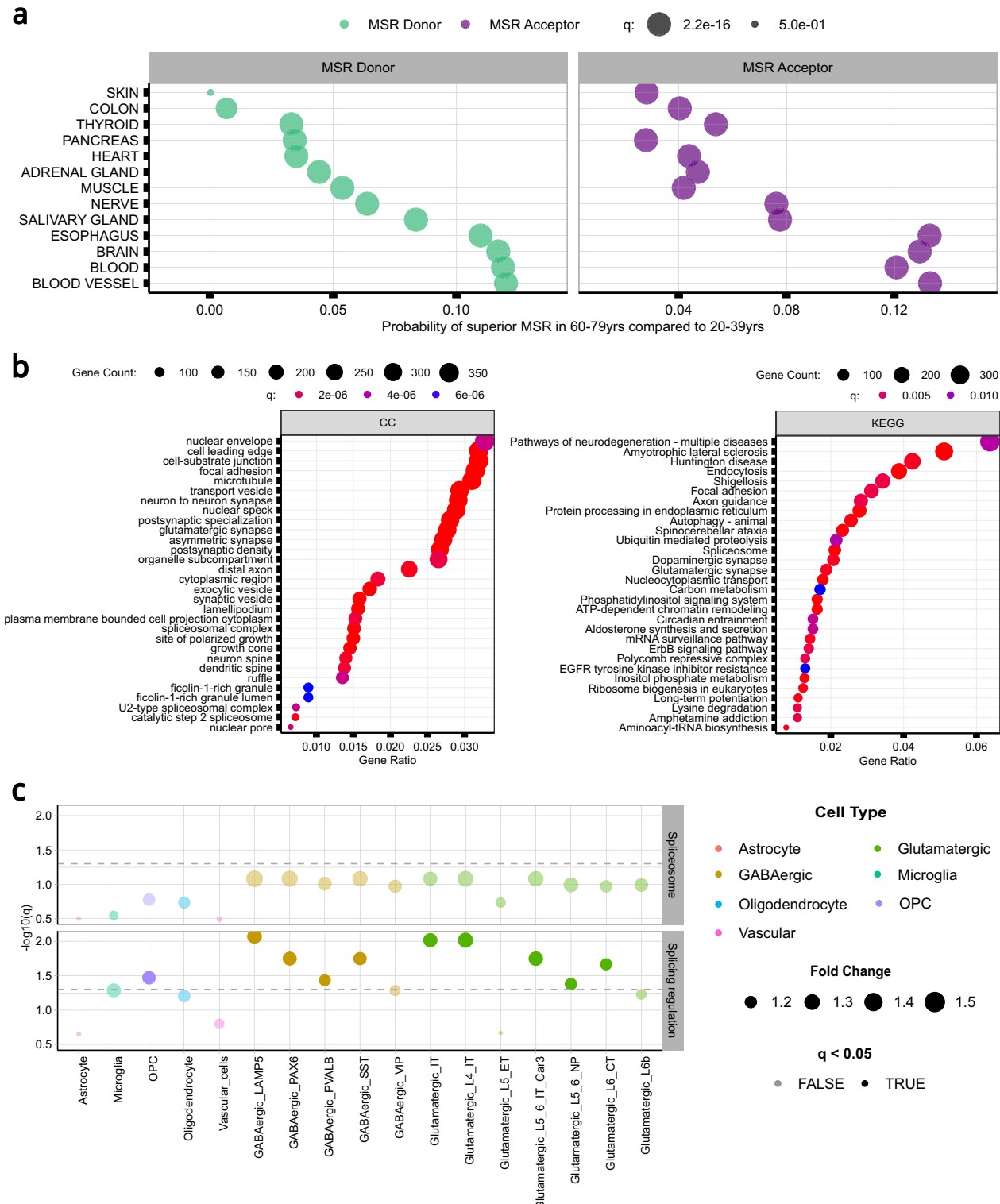

**Fig. 7 | Splicing inaccuracies increase with age and affect genes involved in neuronal function. a** Probability of superior mis-splicing (MSRs) at the 5'ss and 3'ss of the annotated introns in samples from individuals aged between 60-79 years-old as compared to 20-39 yrs. **b** Gene Ontology and KEGG enrichment analysis of the genes containing introns with increasing levels of MSR values with age (i.e. 20-39 yrs < 60-79 yrs) at their 5'ss and/or 3'ss in samples from brain tissues (one-sided over representation analysis test). P-values were corrected for multiple testing using the Benjamini-Hochberg method, resulting in q-values. **c** Cell-type specific expression of 111 splicing-regulator and spliceosomal RBPs (Van Nostrand et al. [45]) in cell types derived from multiple cortical regions of the human brain (Shen et al.[74]). The dashed grey horizontal lines represent the minimum level of significance, with dots displayed above the line showing significant specific expression for a given cell type. P-values were corrected for multiple testing using the Benjamini-Hochberg method, resulting in q-values.

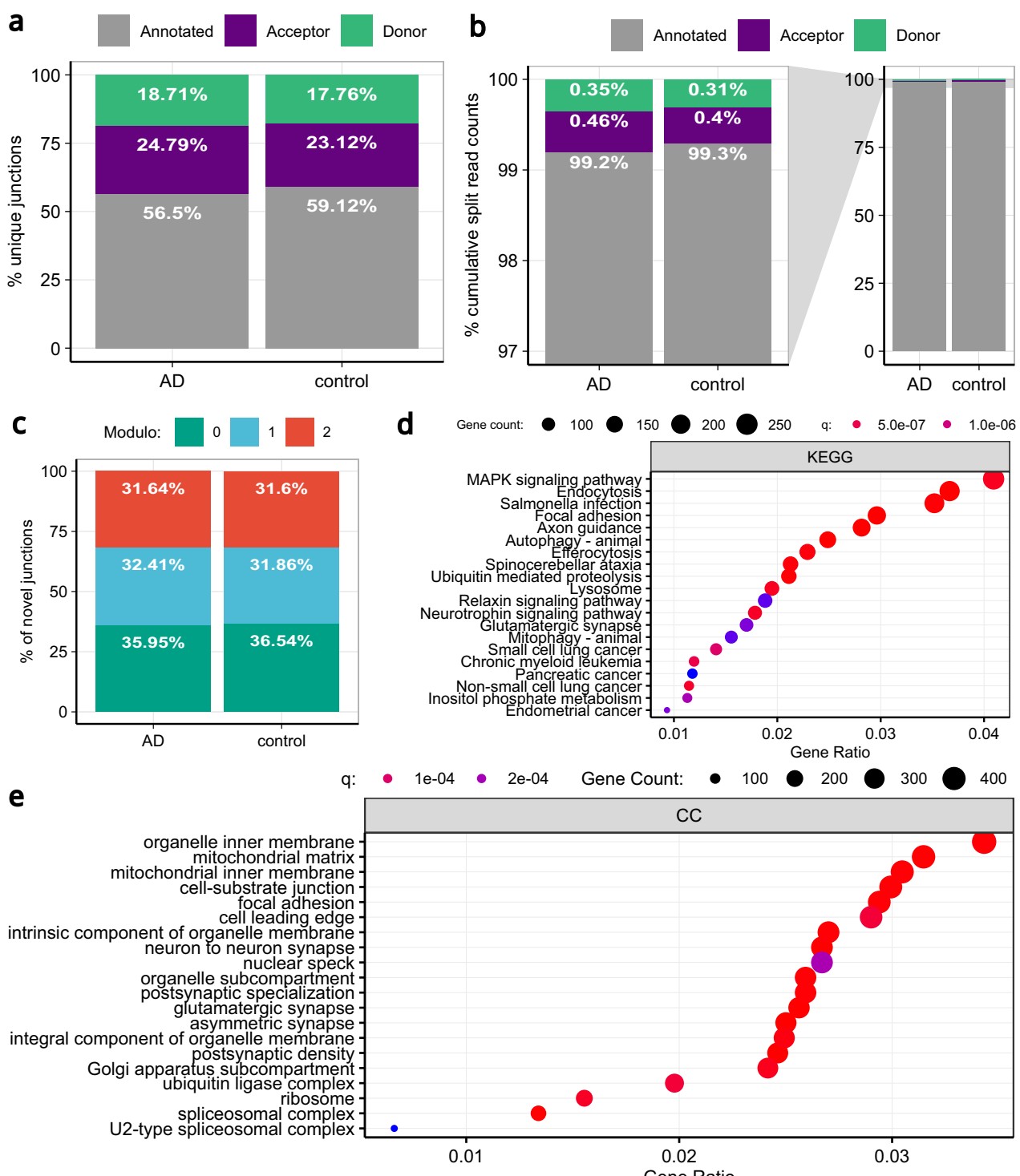

**Fig. 8 | Splicing inaccuracies increase in samples affected with Alzheimer's disease and affect genes involved in synaptic functions. a** Percentage of unique annotated, novel donor and novel acceptor splicing events across AD samples as compared to controls. **b** Percentage of cumulative number of annotated, novel donor and novel acceptor split read counts across AD samples as compared to controls. **c** Percentage of novel junctions that are located at each modulo3 value of the distance to their annotated pairs. **d** KEGG Enrichment analysis of the genes containing introns with higher frequencies of MSRs at any of their two splice sites (i.e. 5'ss and 3'ss) in AD samples as compared to control samples. **e** GO Enrichment analysis of the genes containing introns with higher frequencies of MSRs at any of their two splice sites in AD samples as compared to controls.

inaccuracies in the recognition of splicing signals by the spliceosome machinery itself. We found limited evidence to support the former. While measures of DNA sequence constraint in humans (namely CDTS scores[79]) and local sequence conservation across primates significantly impacted MSRs in all tissues, the effect sizes were variable. When we

compared MSRs in unexposed versus sun-exposed skin (known to have a higher somatic mutation load[65]), we found no significant differences.

These findings are consistent with the current understanding of splicing and its evolution. While splicing is thought to have arisen

through the self-removal of introns from primitive RNA molecules[80], it is postulated that their strict sequence and structural requirements progressively relaxed over time[81]. Consequently, these introns became more reliant on accurate expression of spliceosome RNAs and proteins for efficient recognition of SREs and proper splicing. We suspected that the variable effect of sequence conservation on MSRs across human tissues could be explained by differences in the expression of these components, making splicing inaccuracies primarily a problem of inaccurate sequence recognition.

We formally assessed this hypothesis using publicly available data from the ENCODE consortium to measure MSRs following shRNA knockdown of multiple RBPs[45] and NMD factors. Despite the essential role of *UPF1* and *UPF2* in degrading aberrant transcripts, knocking down these NMD components appeared to lead to a modest increase in splice-site noise, though we recognise the experimental limitations of this analysis. Depending on the RBP targeted, there were distinctive patterns of splicing inaccuracies, suggesting a dependency on adequate levels of expression of each spliceosomal component to accurately target a splice site. Surprisingly, shRNA knockdowns of core spliceosomal molecules, such *AQR* and *U2AF2*, did not reduce the total levels of splicing activity. Instead, these knockdowns appeared to change splice site selection, reducing the overall accuracy of this process. Certainly, mutations in *U2AF* are rate-limiting for splice site choice[82–84]. To support the hypothesis that variability in RBP expression is an important driver of transcriptome-wide splicing accuracy, we co-analysed knockdown data from ENCODE with information on RBP binding sites derived from CLIP-seq data. We found that introns with the highest binding site densities for a given RBP were also the most inaccurately spliced under knockdown conditions of that RBP, indicating a direct relationship between RBP expression and splicing accuracy.

Given that changes in the activity of core spliceosomal components have been linked to ageing[25,42,71,85], we studied changes in MSRs with age in a range of tissues. This analysis revealed an increase in splicing errors in the eldest group across most body sites, including the brain. Focusing on the human brain due to the known importance of RBPs in brain diseases[86,87] and ageing, we noted that core spliceosomal genes and genes involved in synaptic function and proteostasis were affected by age-related changes in splicing accuracy. This could be due to higher requirements for RBP expression in neurons, as suggested by our cell-type specificity analysis. We further explored this possibility by evaluating MSRs in post-mortem brain samples originating from neurologically normal individuals and those with AD. We found a genome-wide increase in MSRs in AD, again affecting genes involved in synaptic functions. Given that cognitive impairment in AD is thought to be driven by synaptic dysfunction and that ageing is the most important risk factor for AD, these findings overall suggest that age- and disease-associated changes in RBP expression could significantly contribute to the pathophysiology of AD. Finally, we analysed the relative contributions of the NMD machinery and RBP factors to splicing inaccuracies in AD and with increasing ageing. We observed that the expression of RNA splicing factors appeared to produce a larger effect. However, more research is required to disentangle the relative contributions of specific RBPs and NMD components. Furthermore, it would be important to use in vitro models and a range of molecular tools to dissect the relationship between these processes and the integrity of transcripts from a given gene.

We note some important limitations of this study. First, all analyses have been performed using bulk RNA-sequencing data. This is likely to impact our assessment of splicing accuracy and its biological impact, potentially leading to an underestimate of its effect on rarer cell types. Second, the analyses performed in this study were based on a strict distinction between split reads that were found in annotation and those that were not, despite the fact that a lack of annotation does not necessarily imply splicing error or non-functionality. Finally, given that short-distance tandem splice sites may produce novel splicing events with important biological functions[88–90], further analyses would be required to distinguish between these regulated novel events and splicing inaccuracies.

Taken together, our results show that inaccurate splicing is common and that understanding its patterns will inform our understanding of the role of splicing integrity in ageing and disease, particularly in the human brain. We believe that this will be key to the successful application of RNA-targeting therapies.

## Methods

### GTEx v8 RNA-sequencing data download and processing

We downloaded and processed data from the IntroVerse database[50], which contains the splicing activity of 332,571 annotated introns (as defined by Ensembl-v105) and a linked set of 1,950,821 novel donor and 2,728,653 novel acceptor junctions, covering 17,510 human control RNA samples and 54 tissues. This dataset of exon-exon junctions was originally provided by the Genotype-Tissue Expression Consortium (GTEx) v8[44] and processed by the recount3[91] (version 1.0.7, https://github.com/LieberInstitute/recount3) project.

The Illumina TruSeq library construction protocol (non-stranded 76 bp-long reads, polyA+ selection) was used in GTEx v8. Samples from GTEx v8 were processed by the recount3 project through Monorail[91] (version 1.0.0, https://github.com/langmead-lab/monorail-external, https://doi.org/10.5281/zenodo.5576208) which uses STAR[92] (RRID:SCR_004463, http://code.google.com/p/rna-star/) to detect and summarise exon-exon splice junctions for each sample. Megadepth[93] (version 1.0.3, RRID:SCR_022779, https://github.com/ChristopherWilks/megadepth) was also used by recount3 to analyse the BAM files output by STAR (version 2.7.3a, RRID:SCR_004463, http://code.google.com/p/rna-star/), with --outSJfilterOverhangMin parameter set to 5 (https://gensoft.pasteur.fr/docs/STAR/2.7.3a/STARmanual.pdf). IntroVerse uses the Bioconductor R package dasper[94] (version 1.4.3, http://www.bioconductor.org/packages/dasper) to annotate the split reads (Ensembl-v105) from GTEx v8 and processed by recount3. Within IntroVerse each novel donor and acceptor junction is first carefully quality-controlled (to ensure that novel junctions could feasibly arise through splicing) and then assigned uniquely to a specific annotated intron. Among the quality-control criteria applied by IntroVerse, all split reads shorter than 25 base pairs (bp) were discarded as well as all split reads located within unplaced sequences on the reference chromosomes and overlapping any of the regions published within the hg38 ENCODE Blacklist[51] (v2.0, https://github.com/Boyle-Lab/Blacklist/blob/master/lists/hg38-blacklist.v2.bed.gz). This 25 bp length filter represents the minimum intron length required for intron splicing (without the inclusion of a portion of either of the two flanking exons). We modified the original structure of the pipeline provided by IntroVerse and added the following data filters. First, samples from fresh frozen preserved tissues were prioritised. On this basis, samples from Brain-Cortex and Brain-Cerebellum tissues were discarded. Second, as all sex-specific tissues and tissues with less than 70 samples (e.g. Bladder, Cells - Leukaemia cell line (CML), Cervix - Ectocervix, Cervix - Endocervix, Fallopian Tube and Kidney - Medulla) were discarded. Third, only samples presenting an RNA Integrity Number (RIN) higher or equal to 6 were included in this study, as any more stringent RIN thresholds would have reduced excessively the number of samples available for study: i) RIN ≥ 8 N Samples Available = 4,127; ii) RIN ≥ 7 N Samples Available = 9,301; iii) RIN ≥ 6 N Samples Available = 13,949. Fourth, we discarded $n = 555$ annotated introns reported to be spliced by the minor spliceosome[52] and $n = 9,252$ novel donor and novel acceptor junctions linked to them. We discarded these minor introns because, even though they represent less than 1% of all intervening sequences in the human genome,

their consensus splicing sequences differ considerably from the consensus sequences of the human introns targeted by the major spliceosome[95]. These filters resulted in a new relational database, namely Splicing intron database, which included a set of 324,956 annotated introns (Ensembl-v105) and a linked set of 3,865,268 novel junctions, originating from 32,026 genes and 201,541 transcripts, and covering 13,949 different human samples and 40 human tissues (Supplementary Fig. 1a,b). All types of exon-exon junction reads were considered (jxn_format=ALL), recount3::create_rse_manual() function (Bioconductor R package recount3 version 1.0.7, https://bioconductor.org/packages/release/bioc/html/recount3.html).

### Calculating the reclassification rates across multiple versions of the Ensembl reference transcriptome

Split reads were first annotated based on the reference transcriptome Ensembl-v97 (v97) released in July 2019 and using the Bioconductor R package *dasper* version 1.4.3 (https://bioconductor.org/packages/release/bioc/html/dasper.html). Per each tissue, we compared the introns that had been classified as novel donor or novel acceptor junctions using v97 but were also re-annotated as annotated introns in the Ensembl-v105 (v105), and used them as a measure of junction re-classification. To create a normalised measure of reclassification rates across the tissues, we divided the number of novel junctions in v97 that had been classified as annotated introns in v105 by the total number of novel junctions that had maintained annotation category between the two aforementioned Ensembl versions.

$$C_T^{v97} = \left( \frac{j}{y} \right) \tag{1}$$

Let $j$ denote the total number of unique novel donor and novel acceptor junctions in v97 that had been re-classified as annotated introns in v105. Let $y$ denote the total number of unique novel donor and novel acceptor junctions in v97 that had maintained annotated category in v105. Let $T$ denote the tissue studied.

This approach was mirrored to reannotate all split reads from the frontal cortex brain tissue using four different Ensembl versions v76, v81, v90 and v104 published in July 2014, July 2015, July 2017 and March 2021, respectively. Reclassification rates in each Ensembl version were again calculated using v105 as the reference annotation.

### Calculating the percentage of unique novel junctions and novel split read counts per tissue

Focusing on the novel donor category, the percentage of unique novel donor junctions in a given tissue was calculated by dividing the cumulative number of unique novel donor junctions across all samples of the studied tissue by the total number of unique annotated introns, novel donor and acceptor junctions found across the same set of samples. Finally, we converted the resulting ratio to a percentage.

$$Pj_T^x = \left( \frac{\Sigma_{i=1}^N x_i}{\Sigma_{i=1}^N x_i + \Sigma_{i=1}^N y_i + \Sigma_{i=1}^N z_i} \right) *100 \tag{2}$$

Let $x$ denote the total number of unique novel donor junctions within one sample of the tissue $T$ studied. Let $y$ denote the total number of unique novel acceptor junctions within one sample of tissue $T$. Let $z$ denote the total number of unique annotated introns within one sample of tissue $T$. Let $N$ denote the total number of samples studied of tissue $T$. Let $T$ denote the tissue studied.

We mirrored the method detailed above to calculate the percentage of unique annotated introns and the percentage of unique novel acceptor junctions within a tissue. Similarly, focusing on the novel donor category, the percentage of novel donor read counts in a given tissue was calculated by dividing the cumulative number of novel

donor reads counts by the total number of reads mapping to annotated introns, novel donor and acceptor junctions across all samples of the tissue studied. The resulting ratio was multiplied by 100 to create a percentage.

$$Pr_T^a = \left( \frac{\Sigma_{i=1}^N a_i}{\Sigma_{i=1}^N a_i + \Sigma_{i=1}^N b_i + \Sigma_{i=1}^N c_i} \right) *100 \tag{3}$$

Let $a$ denote the total number of read counts that all novel donor junctions presented within one sample of tissue $T$. Let $b$ denote the total number of read counts that all novel acceptor junctions presented within one sample of tissue $T$. Let $c$ denote the total number of read counts that all annotated introns presented within one sample of tissue $T$. Let $N$ denote the total number of samples studied of tissue $T$. Let $T$ denote the tissue studied.

We mirrored the formula above to calculate the percentage of annotated introns and novel acceptor read counts within a tissue.

### MaxEntScan score analyses

The MaxEntScan[59] (MES) algorithm (version 1.0, RRID:SCR_016707, http://genes.mit.edu/burgelab/maxent/Xmaxentscan_scoreseq.html) was applied to score the 9 bp sequence at the 5'ss and the 23 bp sequence at the 3'ss of each annotated intron and novel junction stored on each database produced. We downloaded the Human Primary DNA Assembly hg38 (https://ftp.ensembl.org/pub/current_fasta/homo_sapiens/dna/Homo_sapiens.GRCh38.dna.primary_assembly.fa.gz, accessed 01-07-2023) and used the command "samtools faidx Homo_sapiens.GRCh38.dna.primary_assembly.fa" to index the sequence of the hg38 fasta file. Secondly, we obtained the MES software from (http://hollywood.mit.edu/burgelab/software.html, accessed 01-07-2023). Using the indexed huma.primary.assemblyGRCh383.fa file, we extracted the 9 bp and 23 bp motif DNA sequences overlapping the 5'ss and the 3'ss, respectively, of all annotated, novel donor and novel acceptor split reads (Ensembl v105) considered. Next, we used the MaxEntScan software to calculate the scores corresponding to each motif sequence, namely the MES scores. The higher the MES score assigned to a given sequence, the more closely related to a real annotated splice site the sequence is considered.

To investigate the differences in the strength implied by each novel splice site and the analogous annotated splice site of its paired annotated intron, we obtained the delta values of their MES scores. Focusing on the novel donor junctions, the delta MES 5'ss (ΔMES5ss) was calculated by obtaining the difference between the MES score assigned to the 9 bp sequence at the 5'ss of the annotated intron minus the MES score assigned to the 9 bp sequence at its paired 5'ss of the novel donor junction. Similarly, to calculate the delta MES at the acceptor sites (ΔMES3ss), we obtained the difference between the MES score assigned to the 23 bp sequence at the 3'ss of the annotated intron and the MES score assigned to the 23 bp sequence at the 3'ss of its linked novel acceptor junction.

### Calculating the genomic distance and modulo3 values

Per each tissue analysed, we calculated the distances lying between each novel splice site and the analogous annotated splice site of their linked annotated intron. Focusing on the novel donor junctions, we obtained the distances in bp lying between the novel 5'ss of each novel donor junction and the annotated 5'ss of their linked annotated intron. We repeated this process to calculate the distances at 3'ss. Distances in bp were calculated by following a 0-based genomic-interval approach, as we required splicing to occur at precise annotated genomic coordinates to consider splicing as accurate. For instance, focusing in a novel donor junction whose novel 5'ss is located at the *gcNovel* genomic coordinate, the distance lying between *gcNovel* and the 5'ss

of its linked annotated intron *gcIntron* can be expressed as:

$$distance\,(bp) = gcIntron - gcNovel \qquad (4)$$

Let *gcIntron* denote the genomic coordinate corresponding to the 5'ss of the annotated intron *Intron* (Ensembl-v105). Let *gcNovel* denote the genomic coordinate corresponding to the 5'ss of the novel donor *Novel* attached to the annotated intron *Intron*. Let *distance* denote the difference in bp between the two genomic positions *gcIntron* and *gcNovel* within the same strand.

The formula above was mirrored to calculate the distances lying between each novel acceptor junction and its linked annotated intron. For the Modulo3 analysis, we restricted the analysis to the annotated introns belonging to transcripts categorised as MANE Select[96], as these represented exact matches in exonic regions between Refseq transcript and the Ensembl/GENCODE. Only the novel junctions located less than 100 bp apart from annotated splice sites were considered. This filter increased the confidence for the novel products to be located within the adjacent exon and intron sequences, as the average exon size corresponds to 120 bp[97], whereas the mode, median and average length of the annotated introns corresponded to 88 bp, 1,945 bp and 8,388 bp, respectively (Supplementary Fig. 28).

### Calculating the Mis-Splicing Ratio measures

Focusing on the frequency of splicing inaccuracies at the 5'ss of a given annotated intron, the $MSR_D$ measure represent the ratio between the cumulative number of novel donor read counts and annotated read counts linked to the annotated intron of interest detected across all samples of a given tissue.

$$MSR_D^{XT} = \left( \frac{\sum_{i=1}^{N} j_i}{\sum_{i=1}^{N} j_i + \sum_{i=1}^{N} s_i} \right) \qquad (5)$$

Let *j* denote the total number of novel donor junction reads assigned to the annotated intron *X* within one sample of the tissue *T*. Let *s* denote the total number of annotated intron reads for the same intron, *X*, within the same sample of study. Let *N* denote the total number of samples studied from the tissue *T*.

The $MSR_D$ and $MSR_A$ represent bounded measures between [0,1]. Focusing on the $MSR_D$ ratio, $MSR_D = 0$ would represent absence of evidence for splicing inaccuracies at the 5'ss of a given annotated intron, whereas $MSR_D \approx 1$ would represent high mis-splicing activity at the 5'ss of a given annotated intron.

### Calculating the transcript per million measure

Given that poly-adenine (poly-A) selected RNA-sequencing data primarily captures mRNA transcripts with a poly-A tail where splicing has already occurred, hence lacking intronic sequences, the effective length of the gene for read count analyses would be represented by the length of its coding sequence. With this in mind, and to calculate the Transcript Per Million value, namely the TPM measure, per gene, we used the function getTPM() (recount[98] R package, version 1.24.1).

### Using zero-inflated poisson regression models to predict the MSRs

Due to the sparsity of the novel split read data considered in this project, we use a zero-inflated poisson (ZIP) regression to model the genomic characteristics potentially influencing the MSRs at each splice site of the annotated introns studied. We used the zeroinfl function (pscl[99] R package, version 1.5.5.1). As predictors, we included covariates encompassing diverse gene and intron-level features. The gene-level covariates included i) the total number of transcripts of the gene (Ensembl v105) and ii) the percent of protein-coding transcripts in which the assessed intron may appear. The intron-level covariates

included i) the MES[59] scores of the sequences overlapping the 5'ss and 3'ss, ii) the intron length in bp, iii) the mean interspecies conservation score across 17 primate species[100] (phastCons17) and iv) the mean context-dependent tolerance score (CDTS) scores[79] overlapping the proximal intronic sequences. Assuming that cis-acting splicing regulatory sequences primarily lie within 100 bp of exon-intron junctions in the intronic sequence[101], we defined the proximal intronic sequences as the |+100 bp sequence downstream the 5'ss of each annotated intron, and the -100| bp sequence upstream the 3'ss of each annotated intron (| representing the last/first base-pair of the upstream/downstream exon). The mean phastCons17 score represents the probability of negative selection based on the number of substitutions[100] occurring across 17 species (human and 16 primates) during evolution. The CDTS score[79] is a measure to evaluate the sequence constraint of the human population across noncoding regions. This score ranges between negative and positive values, with positive values indicating regions of the human genome which have the highest (i.e. least constrained) sequence variation across humans. Per each tissue analysed, we discarded all annotated introns that were shorter than 200 bp to avoid including overlapping sequences for the conservation and constraint scores included in the model. Prior ZIP model fitting, MSR measures were transformed to integer values. Per tissue, the formula used to build each ZIP model corresponded to:

$$Y = \beta_0 + \beta_1 X_1 + \beta_2 X_2 + \beta_3 X_3 + \beta_4 X_4 + \beta_5 X_5 + \beta_6 X_6 + \beta_7 X_7 + \beta_8 X_8 + \beta_9 X_9 + \varepsilon_0$$
$$(6)$$

where the dependent variable corresponded to: Y = MSR ($MSR_D$ or $MSR_A$) of each annotated intron, transformed to an integer value using the constant 100,000. The independent variables corresponded to: $X_1$=number of transcripts in annotation of the gene; $X_2$=percentage of protein coding transcripts in which the intron may appear; $X_3$ = MES of the 5'ss of the intron; $X_4$ = MES of the 3'ss of the intron; $X_5$=Mean PhastCons17 score of the 100 bp sequence downstream the 5'ss of the intron; $X_6$=Mean PhastCons17 score of the 100 bp sequence upstream the 3'ss of the intron; $X_7$=Mean CDTS score of the 100 bp sequence downstream the 5'ss of the intron; $X_8$=Mean CDTS score of the 100 bp sequence downstream the 3'ss of the intron; $X_9$=intron length in bp; $0 = N(0, sigma^2)$.

In total, 80 ZIP models were generated, corresponding to two ZIP models per GTEx tissue considered (40 tissues and two MSR measures per tissue evaluated, the $MSR_D$ and the $MSR_A$). For robust standard error calculation of the beta coefficients, we used the coeftest(vcov = sandwich::sandwich) function (R package lmtest[102], version 0.9-40, https://cran.r-project.org/web/packages/lmtest/lmtest.pdf). P-values across the 80 models generated were FDR-adjusted. Beta coefficients generated by each ZIP model produced in each tissue were grouped by covariate, generating a distribution of beta coefficients across tissues. Prior results visualisation, beta coefficients were transformed from log scale to exponential values, creating exponentiated beta coefficients and indicating the multiplicative effect on the MSR for a 1-unit increase per each independent variable. Beta coefficients were not further adjusted due to the scaling of the dependent variable by 100,000, as this transformation affects the scale of the outcome rather than the interpretation of individual coefficients. Covariates were not centred prior to model fitting in any of the ZIP models produced.

### Assessing the levels of MSRs in sun-exposed versus not-sun-exposed skin tissues

We selected all annotated introns from the Skin - Sun Exposed (Lower leg) and the Skin - Not Sun Exposed (Suprapubic) body sites, and evaluated their differences in MSRs at their 5'ss ($MSR_D$) and 3'ss ($MSR_A$). We obtained the common annotated introns overlapping both tissues ($n = 245,349$). In addition, to reduce any potential biases derived from differences in the sequencing depth levels of the two sets

of samples, we only kept the common annotated introns with similar expression levels between the two body sites by restricting the maximum difference in log10 mean expression to 0.005 reads (matchit() function, MatchIt R package[103], version 4.4.0, https://cran.r-project.org/web/packages/MatchIt/vignettes/MatchIt.html). Finally, we obtained the $MSR_D$ and $MSR_A$ values from each of the $n = 245,349$ annotated introns of either Skin - Sun Exposed (Lower leg) and Skin - Not Sun Exposed (Suprapubic). To test for any significant differences in the median distribution of these two MSR measures between the two skin body sites, we used a one-tailed paired Wilcoxon signed rank test function with continuity correction (wilcox_test() function, R package rstatix[104] version 0.7.1, RRID:SCR_021240, https://CRAN.R-project.org/package=rstatix).

### Analysing shRNA knockdown of RBPs followed by RNA-sequencing data from ENCODE

From the list of 356 RBPs published by Nostrand et al. in ref. 45, we selected 115 RBPs that had been functionally categorised as splicing regulation, spliceosome or EJC by the authors. We also downloaded a second list of 118 human genes published by the Reactome project that had been classified as involved in NMD processes [R-HSA-927802, NMDv3.7, Browser v82], which included *UPF1* and *UPF2* due to their known importance in NMD. As a control gene, we selected *SAFB2*, a gene coding for an RBP with no known impact on splicing, spliceosome structure, EJC identification or NMD[45]. In total, 235 genes were considered for study. From the 235 genes initially considered, only 54 had 8 shRNA knockdown followed by RNA-sequencing data experiments available on the ENCODE platform. A total of 8 alignment BAM files (GRCh38 v29) were downloaded per gene, each one corresponding to a different ENCODE experiment. Experiments were chosen based on similarity of metadata and design. Briefly from ENCODE: i) 4 experiments with RNA-sequencing data available on K562 and HepG2 cells treated with an shRNA knockdown against a given gene, and ii) 4 control shRNA experiments against no target gene were chosen for each gene. To extract the splicing junctions from the BAM files to a BED12 format, we made use of the command "junction extract" made available through the regtools software package (version 0.5.2, http://regtools.org/). We required i) a minimum anchor length of 8 bp and ii) a minimum and maximum intron size of 25 and 1,000,000 bp, respectively, to call the presence of a junction. The strand information was provided by the aligner. Prior to the extraction, alignment reads were sorted and indexed using the commands sort and index, both made available through the SAMTOOLS[105] software (version 1.16.1, RRID:SCR_002105, http://htslib.org/). We then applied a similar data analysis to the one originally published by IntroVerse, and created a separate database for each ENCODE shRNA knockdown project, in which samples were clustered following a case/control grouping criteria. Case samples corresponded to the experiments in which a gene had been targeted for knockdown, whereas control samples corresponded to untreated controls in which no gene had been targeted. This database stored splicing information about $n = 276,589$ unique annotated introns (Ensembl-v105) that were found across the 4 shRNA knockdown and 4 control experiments studied per each of the 54 RBPs evaluated. From the 276,589 annotated introns stored, 163,099 presented evidence of at least one type of novel donor or novel acceptor splicing event. It also included 344,713 novel donor and 617,016 novel acceptor junctions, covering 185,022 transcripts, 25,578 genes and 432 ENCODE experiments. To account for any differences in read-depth or RIN numbers across the different samples and experiments compared, we only considered the annotated introns that were common across all experiments. For more details about how the MES, distances, modulo3 and MSR measures were calculated, please refer to the corresponding Methods sections in this manuscript. To detect any significant differences for each gene between case versus control samples in the MSR values of the common introns across experiments, we made use of the wilcox_test function (R package rstatix[104] version 0.7.1, RRID:SCR_021240, https://CRAN.R-project.org/package=rstatix). A total of 116 one-tailed Wilcoxon tests were run, one per ENCODE knockdown project and splice site. The p-values obtained from each test were adjusted using the Bonferroni correction method. In those cases in which the alternative hypothesis ($H_1$) was accepted, we calculated the probability of superior MSR outcome in case vs control samples by using the function wilcox_effsize() (R package rstatix, version 0.7.1, RRID:SCR_021240, https://CRAN.R-project.org/package=rstatix).

### ENCODE shRNA knockdown efficiency extraction

To obtain a measurement of the knockdown efficiency for each ENCODE experiment, we identified a biosample preparation and characterization document attached to 46 out of the 54 studied genes. The efficiency is calculated by comparing protein levels in control and knockdown cells using a western blot analysis, and reported in figures embedded in the document. To extract the figures, we made use of the fitz module available from the python package PyMuPDF[106] (version 1.21.1, https://github.com/pymupdf/PyMuPDF). We employed the Tesseract-OCR (Optical Character Recognition) algorithm, available through the python package pytesseract (version 0.3.10, https://pypi.org/project/pytesseract/) to extract the text from the images. To ensure high accuracy in the image to text conversion, figures were: (1) cropped to only contain the depletion percentages, and (2) resized to a lower resolution to better match the training data of the OCR algorithm. No additional configuration was specified to the Tesseract-OCR engine. A perfect accuracy was observed when tested in 15% of the samples, and outliers were manually verified. The final reported knockdown efficiency is the average of the measurements for all four samples.

### Analysis of RBP-RNA interactions using CLIP-seq data

Given the evidence indicating that RBP expression is likely to be cell-type specific[45], we linked ENCODE RBP knockdown data with RBP-RNA interactions supported by the binding sites of RBPs derived from CLIP-seq from the ENCORI platform[46], as both sources of data had been created from K562 and HepG2 cell lines. Of the 54 RBPs considered within the shRNA knockdown analysis, only 15 RBPs related to splicing regulation and spliceosome assembly had CLIP-seq data available for HepG2 and K562 cell lines on the starBase/ENCORI platform. To download data, we used the available API (https://rnasysu.com/encori/tutorialAPI.php, accessed 03/04/2024, Assembly=hg38, GeneType=mRNA, RBP=name of each RBP, clipExpNum=1, pancancerNum=0, target=all, cellType=all). For each of the 15 RBPs considered, we obtained the annotated introns with increasing MSR levels, either at their donor or acceptor splice sites, in samples under knockdown conditions of each RBP, namely case samples, and compared with untreated control samples. We built a contingency table using the number of introns displaying higher/lower levels of MSR at any of their two splice sites in case samples and the number of introns with binding sites for the studied RBP either within their intronic sequence or in close proximity of their donor and acceptor boundaries (-/+ 100 bp). We performed a chi-square test (function chisq.test, R package stats, version 4.0.5, RRID:SCR_025968, https://stat.ethz.ch/R-manual/R-devel/library/stats/html/00Index.html). The null hypothesis tested per RBP studied corresponded to: "$H_0$: There is no significant difference in MSR changes between the annotated introns from knockdown versus control experiments and the number of binding sites that the intron presents for that RBP". We ran a chi-square test per MSR measure.

### RBP expression levels across tissues

We visualised the gene expression for 115 important spliceosomal RBP genes across 42 GTEx v8 tissues, deriving the RBP gene list from Van

Nostrand et al. [45]. In order to gauge cross-tissue variation in expression for each gene, the following calculations were performed on a per-gene basis. Firstly, we obtained the cross-sample expression for each tissue, identifying the tissue with the median expression value, namely tissue Y. Next, we calculated the log2 fold-change in expression for each of the remaining 41 tissues in relation to expression in tissue Y. Finally, the log2 fold-change expression values for each gene were visualised as a heatmap facetted by gene functional groups. The functional categories used were Splicing Regulation, Spliceosomal and EJC, obtained from Van Nostrand et al. [45]. The code to reproduce this analysis can be accessed at https://github.com/ainefairbrother/RBP_expression_analysis (version 1.0.0, https://doi.org/10.5281/zenodo.7736907).

### Changes in RBP expression levels with age

We downloaded raw read counts from all genes expressed within each of the GTEx v8 tissues available on the recount3 project. We used the function create_rse_manual() (R package recount3, version 1.0.7, https://bioconductor.org/packages/release/bioc/html/recount3.html). Raw counts were transformed using the function transform_counts() (R package recount3, version 1.0.7, https://bioconductor.org/packages/release/bioc/html/recount3.html). To calculate the gene expression within each sample we obtained its corresponding Transcript per Million (TPM) value. TPM data was used in this analysis because all samples had been obtained from the same tissue each time, and all samples had been sequenced using the same library protocol, polyA-selection, reducing the risk of misleading TPM comparisons[107]. To know whether the expression levels of the 116 RBPs involved in splicing regulation, spliceosomal and EJC recognition[45] and the 5 NMD genes studied were affected by age across tissues, we built a linear regression model per RBP. The independent variable to predict corresponded to the TPM value in log10 scale of each RBP in each sample. The dependent variables corresponded to a set of covariates providing information about the sample: age, center, gebtch, gebtchd, nabtc, nabtchd, nabtcht, hhrdy, sex and rin. These covariates were chosen on the basis of the principal component analysis (PCA) results published by Fairbrother-Browne, A. et al. [108] using data from GTEx v6. Some of these covariates were categorical, so we transformed them into numerical values prior inclusion to the linear models. In total, 121 linear models were run per GTEx tissue, one linear model per 116 RBPs and 5 NMD genes studied. Each linear model was built to predict N TPM values per RBP, with N equating to the total number of samples available per tissue. P-values produced by each linear model were corrected for multiple testing using the Benjamini-Hochberg method, producing q values. Finally, in those cases in which the age covariate produced a negative estimate value in the prediction of the TPM for a given RBP, it was considered that age negatively affected the expression levels of that given RBP across the set of samples studied.

### Age stratification and sample clustering

GTEx samples were grouped by tissue following the original classification made by recount3[91] (Supplementary Table 14). Samples from each body region were then binned by age within one of these three categories 20-39, 40-59 and 60-79 years-old. Only the body sites presenting a minimum of 75 samples, equating to at least 25 samples per age category, were considered. These were 18 body sites in total: ADIPOSE TISSUE, ADRENAL GLAND, BLOOD, BLOOD VESSEL, BRAIN, COLON, OESOPHAGUS, HEART, LUNG, MUSCLE, NERVE, PANCREAS, SALIVARY GLAND, SKIN, SMALL INTESTINE, SPLEEN, STOMACH and THYROID. To account for differences in the RIN numbers presented by the samples grouped in each age category, we down sampled the clusters 40-59 and 60-79 to meet similarity with the 20-39 group, as the overall sample size of the latter was always lower than the two former categories across all body sites studied. The sample pairing was performed only when two

samples from each age group presented a maximum difference of 0.05 in their RIN numbers (matchit(), MatchIt R package[103], version 4.4.0, https://cran.r-project.org/web/packages/MatchIt/vignettes/MatchIt.html) (Supplementary Fig. 19). We then applied our modified version of the pipeline published by IntroVerse and created a relational database to study the changes occurring in the splicing activity of the $n = 321,663$ annotated introns (Ensembl-v105) that were found across the three age categories and 18 body sites studied. We named it the Age-Stratification intron database (Supplementary Fig. 20). From the 321,663 annotated introns stored, 254,416 presented evidence of at least one type of MSR event. It also included 1,183,988 novel donor and 1,664,788 novel acceptor junctions, covering 200,837 transcripts, 31,544 genes and 6519 samples from 40 body sites and 18 tissues. To study the effect size of MSR produced by age at the 5'ss and 3'ss of the annotated introns stored on the Age-Stratification intron database, we made use of their $MSR_D$ and $MSR_A$ values. To reduce any biases in the number of annotated introns considered in this comparative analysis across multiple body sites, we only included the introns that were common across the three age categories and all 18 tissues studied. These were a total of $n = 112,523$ common annotated introns. Then, to further reduce the likelihood of including borderline samples between the three age groups, we only considered samples from the two most extreme age clusters 20-39 and 60-79. Focusing on the 5'ss of the $n = 112,523$ common annotated introns overlapping the 20-39 and 60-79 age groups, we calculated the Wilcoxon effect size that the covariate age (i.e. 20-39 and 60-79) produced over their $MSR_D$ values. We then repeated this approach to measure the effect size between age and the MSR at the 3'ss ($MSR_A$) of the same set of $n = 112,523$ annotated introns. In both cases, we made use of the function wilcox_effsize (R package rstatix[104], version 0.7.0, RRID:SCR_021240, https://CRAN.R-project.org/package=rstatix). All Wilcoxon tests were performed using a one-tailed test. Next, we measured MSR differences with age after controlling for RBP and NMD expression. To do this, we calculated the fold-change in TPM expression of each RBP/NMD factor in the 60-79 as compared with the 20-39 years-old group, and we normalised the MSR values on the basis of the inverse fold-change. Samples within the two clusters had been previously subsampled to meet by RIN similarity. We then repeated the assessment of age differences in MSR values between the two age groups as previously described by using the Wilcoxon Effect Size method (function wilcox_effsize, R package rstatix[104], version 0.7.0, RRID:SCR_021240, https://CRAN.R-project.org/package=rstatix). Finally, to assess the impact of each RBP/NMD factor on MSR values, we ranked each of the factors in terms of their contribution to age-related splicing inaccuracies.

### GO and KEGG enrichment analyses of genes containing introns with increasing levels of MSR values in ageing samples

Using data from the Age-Stratification database, we selected all introns overlapping the three age categories for the brain tissue. These were $n = 211,178$ annotated introns. To assess any changes occurring in their splicing activity, we compared their $MSR_D$ and $MSR_A$ measures and evaluated the changes occurring as the age of each cluster increased. Focusing on the $MSR_D$ value, we selected the introns presenting increasing levels of MSR with age at their 5'ss ($MSR_D$ 20-39 < 40-59 < 60-79 yrs). We mirrored this approach focusing on their $MSR_A$. Then, we obtained the gene symbol of all introns showing increasing $MSR_D$ and/or $MSR_A$ values with age. These were a total of $n = 12,408$ unique genes. Using as background the list of all genes ($n = 20,472$) parenting the complete set of annotated introns found across brain sites, we ran a GO and KEGG enrichment analysis of the set of $n = 8,117$ unique genes. For the GO enrichment analysis, we used the R function enrichGO (R package clusterProfiler, version 3.18.1, RRID:SCR_016884, http://yulab-smu.top/biomedical-knowledge-mining-book/clusterprofiler-go.html). For the KEGG enrichment analysis, we used the R function

enrichKEGG (R package clusterProfiler, version 3.18.1, RRID:SCR_016884, http://yulab-smu.top/biomedical-knowledge-mining-book/clusterprofiler-kegg.html?q=enrichKEGG#clusterprofiler-kegg-pathway-ora).

## RBP cell-type enrichment calculation

We used Expression Weighted Cell Type Enrichment (EWCE)[109] (https://bioconductor.org/packages/EWCE) to determine whether genes involved in splicing regulation have higher expression within particular brain-related cell types than would be expected by chance. We used two gene lists: i) a list of 115 RBPs that had been functionally categorised as splicing regulation, spliceosome or EJC by Nostrand et al. [45], and ii) a list of 118 human genes published by the Reactome project that had been classified as involved in NMD processes [R-HSA-927802, NMDv3.7, Browser v82]. In total, 233 genes were considered for study. Our aim was to evaluate the average level of expression of those 233 genes within the Human Multiple Cortical Areas SMART-seq data set, which includes single-nucleus transcriptomes from 49,495 nuclei across multiple human cortical areas. These data are freely available through the Allen Brain Atlas[74] data portal (https://portal.brain-map.org/atlases-and-data/rnaseq). To achieve this aim, we first downloaded the EWCE docker image (https://hub.docker.com/r/neurogenomicslab/ewce), which includes the EWCE[110] R package (version 0.99.3, https://bioconductor.org/packages/release/bioc/html/EWCE.html). Second, we downloaded the single-nucleus transcriptomes from 49,495 nuclei across multiple human cortical areas from https://portal.brain-map.org/atlases-and-data/rnaseq/human-multiple-cortical-areas-smart-seq. We made use of the matrices including exon and intron counts. For this analysis, all brain regions sampled were included, which corresponded to: Middle temporal gyrus (MTG); Anterior cingulate cortex (ACC; also known as the ventral division of medial prefrontal cortex, A24); Primary visual cortex (V1C); Primary motor cortex (M1C) - upper (ul) and lower (lm) limb regions; Primary somatosensory cortex (S1C) - upper (ul) and lower (lm) regions; Primary auditory cortex (A1C).

Then, we generated the cell type annotations. Level 1) Allen Brain Atlas provided a class and subclass label. Class had only 3 levels (GABAergic, glutamatergic and non-neuronal), thus instead we used the subclass label, which subdivided glutamatergic neurons into 7 subtypes, GABAergic neurons into 5 subtypes, and non-neuronal cell types into Astrocyte, Endothelial, Microglia, Oligodendrocyte, OPC, Pericyte, VLMC. As the number of endothelial cells ($n = 70$), pericytes ($n = 32$) and VLMC ($n = 11$) nuclei was low, these were merged into the class Vascular Cell. Level 2) used the original clusters defined by the Allen Brain Atlas. A total of 1,985 nuclei were labelled as Outlier Calls and were removed during generation of the cell type dataset. We used the function fix_bad_hgnc_symbols() (R package EWCE, version 0.99.3, https://bioconductor.org/packages/release/bioc/html/EWCE.html) to remove any symbols from the gene-cell matrix that were not official HGNC symbols. A total of 30,792 genes were retained. We then used the function drop_uninformative_genes() (R package EWCE, version 0.99.3, https://bioconductor.org/packages/release/bioc/html/EWCE.html), which removes informatic genes to reduce compute time in subsequent steps. The following steps were performed: 1) Drop non-expressed genes ($n = 1,263$). This step removed the genes that are not expressed across any cell types; 2) Drop non-differentially expressed genes ($n = 6,304$), which removes genes that are not significantly differentially expressed across level 2 cell types with an adjusted p-value threshold of 1e-05. Finally, we used the function generate_celltype_data() from the R package EWCE (version 0.99.3, https://bioconductor.org/packages/release/bioc/html/EWCE.html) to generate the celltype dataset. This dataset can be accessed at: https://github.com/RHReynolds/MarkerGenes (version 0.99.1, DOI: 10.5281/zenodo.6418604). In a separate analysis run in R 4.2.0 (https://cran.r-project.org/bin/windows/base/old/4.2.0/), we used this cell type data reference in EWCE. The goal of this analysis was to determine whether the genes of interest had significantly higher expression in certain cell types than might be expected by chance. Bootstrap gene lists controlled for transcript length and GC-content were generated with EWCE iteratively ($n = 10,000$) using bootstrap_enrichment_test() function (EWCE[109] R package, version 1.4.0). In brief, this function takes the inquiry gene list and a single cell type transcriptome data set and determines the probability of enrichment of this list in a given cell type when compared to the gene expression of bootstrapped gene lists; the probability of enrichment and fold-change of enrichment are the returns. P-values were corrected for multiple testing using the Benjamini-Hochberg method. The code, plotting and library versions used for this analysis can be accessed at: https://github.com/mgrantpeters/RBP_EWCE_analysis (version 1.0, DOI: 10.5281/zenodo.7734035).

## Alzheimer's disease/control short-read RNA-sequencing data download and processing

We downloaded from recount3[91] junction data corresponding to 98 fusiform gyrus samples originating from individuals with Alzheimer's (AD) and neurologically normal (control) individuals, which were originally published by Friedman et al. [49] (Gene Expression Omnibus: GSE95587)(recount3 project ID = SRP100948). RNA was extracted from frozen fusiform gyrus tissue blocks of autopsy-confirmed Alzheimer's cases and neurologically normal age-matched controls. Standard polyA-selected Illumina RNA-seq was performed. Only samples with RNA integrity scores of at least 5 as well as post-mortem intervals lower than 5 hr were used. We classified the 98 samples by diagnosis, namely Control and AD groups. Since the presence of split reads within a sample can be affected by the sequencing depth of the sample, we subsampled both sets of samples to match them by mapped read depth similarity. This reduced both sets to 24 samples each. We next built a database following the methods indicated in the present manuscript. This database included a set of 245,738 annotated introns (Ensembl-v105, 149,649 of them with no evidence of mis-splicing and 96,089 introns with at least one linked novel split read), and a linked set of 219,658 novel junctions (125,085 novel acceptor and 94,573 novel donor junctions), originating from 23,999 genes and 181,284 transcripts (Supplementary Fig. 25 and Supplementary Fig. 26a-d). To compare differences in splicing accuracy between the annotated introns found across the 48 samples studied, we made use of their MSR values. To avoid potential biases derived from differences in mean expression levels, we only considered those annotated introns overlapping both groups of samples that displayed a maximum difference in their log10 expression levels of 0.005 (matchit() function, MatchIt R package, version 4.4.0, https://cran.r-project.org/web/packages/MatchIt/vignettes/MatchIt.html). Mean expression levels were measured by obtaining the average number of split reads supporting the presence of each annotated intron across all samples of each group. This subsampling process reduced both distributions of introns to 193,487 in each sample category (Supplementary Fig. 26e,f). In all downstream statistical tests performed, we used one-tailed paired Wilcoxon tests to evaluate differences in the distribution of MSRs at the donor and acceptor (i.e. $MSR_D$ and $MSR_A$) values between the annotated introns overlapping the AD and control sample groups. We measured MSR differences in AD as compared to control samples after controlling for RBP and NMD expression. To do this, we calculated the fold-change in TPM expression of each RBP/NMD factor in AD as compared with the control cluster, and normalised MSR values on the basis of the inverse fold-change. Samples within the two clusters had been previously subsampled to meet by RIN and sequencing depth similarity. We then repeated the assessment of AD/control differences in MSR values between the two disease

groups as previously described by using the Wilcoxon Effect Size method (function wilcox_effsize, R package rstatix[104], version 0.7.0, RRID:SCR_021240, https://CRAN.R-project.org/package=rstatix). Finally, to assess the impact of each RBP/NMD factor on MSR values, we ranked each of the factors in terms of their contribution to AD-related splicing errors (i.e. AD-related effect size).

### Reporting summary

Further information on research design is available in the Nature Portfolio Reporting Summary linked to this article.

## Data availability

This manuscript processes publicly available RNA-sequencing data that is already in the public domain. To download each of the RNA-seq datasets studied, we used the recount3 R package (version 1.0.7, https://github.com/LieberInstitute/recount3), to download 1) data corresponding to the GTEx v8 project (Supplementary Table 14); and 2) projectID = "SRP100948" (Gene Expression Omnibus: GSE95587). Bam files from the ENCODE Gene Silencing Series were downloaded using the R code https://github.com/SoniaRuiz/recount3-database-project/ (https://doi.org/10.5281/zenodo.14204939, v2.0.0), R script '29_ENCODE_download_bams.R', which was adapted from: https://github.com/guillermo1996/ENCODE_Metadata_Extraction (version 1.0.2, https://doi.org/10.5281/zenodo.7733986). The five SQL databases (Splicing, Splicing-2-reads, Age-stratification, Encode-shRNA and AD-control) described in this manuscript, which are generated from the publicly available RNA-seq datasets described above, are available for data download at https://zenodo.org/records/14307072 (https://doi.org/10.5281/zenodo.14307072) and https://rytenlab.com/browser/app/splicing_accuracy_manuscript_databases. Supplementary Tables are available at https://zenodo.org/records/14307072 (https://doi.org/10.5281/zenodo.14307072).

## Code availability

The repositories https://github.com/SoniaRuiz/recount3-database-project (version 2.0.0, https://doi.org/10.5281/zenodo.14204939) and https://github.com/SoniaRuiz/splicing-accuracy-manuscript (version 2.0.0, https://doi.org/10.5281/zenodo.14204490) contain the code (1) to generate the five sqlite databases described in this manuscript and (2) to replicate all analyses, figures, tables and supplementary information included in this manuscript, respectively. All analyses were performed in R version 4.0.2 (https://cran.r-project.org/bin/windows/base/old/4.0.2/) (Ubuntu 16.04.7 LTS). The code used to obtain the metadata and extract the bam files associated with each ENCODE shRNA knockdown data was adapted from https://github.com/guillermo1996/ENCODE_Metadata_Extraction (version 1.0.2, https://doi.org/10.5281/zenodo.7733986) and https://github.com/guillermo1996/ENCODE_Splicing_Analysis (version 1.0.1, https://doi.org/10.5281/zenodo.7733984). The code to calculate the expression levels of the RBPs known to contribute to splicing and its regulation across body sites can be accessed at https://github.com/ainefairbrother/RBP_expression_analysis (version 1.0.0, https://doi.org/10.5281/zenodo.7736907). The code to reproduce the cell type specificity analysis of the set of RBPs known to contribute to splicing and its regulation, and using as reference the drop-seq data from multiple cortical regions (Allen Brain Atlas) is available at: https://github.com/mgrantpeters/RBP_EWCE_analysis (version 1.0, https://doi.org/10.5281/zenodo.7734035). The code to generate the cell type dataset using the function generate_celltype_data() from the R package EWCE, can be accessed at: https://github.com/RHReynolds/MarkerGenes (version 0.99.1, https://doi.org/10.5281/zenodo.6418604).

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

## Acknowledgements

This research was funded in whole or in part by Aligning Science Across Parkinson's [Grant numbers: ASAP-000478, ASAP-000509, and ASAP-000486] through the Michael J. Fox Foundation for Parkinson's Research (MJFF). For the purpose of open access, the author has applied a CC BY public copyright licence to all Author Accepted Manuscripts arising from this submission. S.G.R. and M.R. was supported through the award of a Tenure Track Medical Research Council (MRC) Clinician Scientist Fellowship (MR/N008324/1). E.K.G. was supported by the Postdoctoral Fellowship Program in Alzheimer's Disease Research from the BrightFocus Foundation (Award Number: A2021009F). A.F.-B. was supported through the award of a Biotechnology and Biological Sciences Research Council (BBSRC UK) London Interdisciplinary Doctoral Fellowship; Z.C. was supported by a clinical research fellowship from the Leonard Wolfson Foundation; A.L.G.-M. was supported by Fundación Séneca [21230/PD/19]; J.B. was supported through the Science and Technology Agency, Séneca Foundation, CARM, Spain (research project 00007/COVI/20); L.C.-T. was supported by the National Institutes of Health (United States) [R01MH123567]. D.C.R. was supported by the Michael J. Fox Foundation - ASAP program (ASAP-000486).

## Author contributions

Conceptualization: S.G-R., M.R., D.Z., and S.G. Investigation: S.G-R, M.R. Formal analysis: S.G-R., G.R-P., M.G-P., A.F-B., R.H.R. Figure design and conceptualization: S.G-R., E.K.G., M.R. Writing – original draft: S.G-R., M.R. Writing – review and editing: S.G-R., E.K.G., R.H.R., J.W.B., M.G-P., A.F-B., A.G-M., Z.C., G.R-P. and L.C.T. Funding acquisition: M.R. Supervision: J.B., D.R., L.C.T., and M.R.

## Competing interests

S.G. is a current employee of Verge Genomics. All work performed for this publication was performed in his own time, and not as a part of his duties as an employee. R.H.R and D.Z are current employees of CoSyne Therapeutics. All work performed for this publication was performed in their own time, and not as a part of their duties as employees. The remaining authors declare no competing interests.
