## [Transparent Peer Review file · Nature Communications]

Splicing accuracy varies across human introns, tissues, age and disease

Corresponding Author: Professor Mina Ryten

Version 0:

Reviewer comments:

Reviewer #1

(Remarks to the Author)

This manuscript reports a careful computational analysis of RNA splicing across human tissues, relying merely on RNA-seq data that already exist in public data bases. The results are sound and the figures are creative and well thought through. The manuscript is well written and structured. However, the study does not contribute a significant advance to the splicing field. It has long been established that alternative splicing (which includes most of the reported "mis-spliced" events, except for cryptic splicing caused by DNA mutations) is tissue-specific, and this involves the differential expression/activity of splicing regulatory proteins. Additionally, a major caveat of this study concerns lack of evidence for biological relevance of the findings. Indeed, many of the "mis-spliced" transcripts represent a minor proportion compared to normally processed transcripts. Thus, they may result from splicing noise and have no physiological consequence either for ageing or for disease.

MAJOR COMMENTS:

- 1) It would be good to see an analysis not just of the prevalence of mis-splicing events in different samples but also of the extent to which the specific events are shared between samples. It would help assess to what extent the variability in mis-splicing patterns between samples is quantitative vs qualitative.
- 2) When running MaxEntScan scores, it's unclear whether authors used the genomic sequence of the reference genome, or the one inferred from the genomic data of GTEx. In case of the former, authors should note that the reference genome for those cases is not representative of the biologic conditions where that mis-splicing was generated. Splice sites can be straighten or weaken by the presence of variants that are not currently being considered.
- 3) The authors state that "sequence conservation in genomic regions flanking the 5' and 3' splice sites had the largest effect on splicing accuracy". This cannot be true, at least in a mechanistic sense. (Note that, in accordance with Figure 1b, I am interpreting this statement to be not about how well a given splice site matches the splice site consensus but rather about interspecies conservation - in the figure, the human splice site sequences are identical for the two examples.) The splicing machinery does not know anything about what the sequence is in other species. The conservation of the splice sites is merely an indicator that these introns are functionally important, and so it makes sense that there would be a pressure to splice them accurately. But the actual mechanistic determinants of why the splicing is more or less accurate for a given intron must be elsewhere - in cis-regulatory elements, in chromatin structure, in the exon-intron architecture... Both for the phastCONS as well as the CDTs analysis, the authors use very short sequence regions that would not necessarily contain all the relevant cis-regulatory information. It would be interesting to include conservation/polymorphism levels within these larger regions as predictors in the regression models to see if perhaps the coefficient for splice site conservation decreases.
- 4) A critical issue is how much mis-splicing is missed because of NMD. The authors do two analyses that partially address this point but both also have alternative interpretations. Firstly, the 3-nucleotide periodicity of the novel splice junctions in coding regions could reflect that the other mis-splicing events get gnawed up by NMD - or it could reflect some property of the genetic code, perhaps even some sort of evolved robustness against splicing errors. Secondly, the fact that non-coding genes show more mis-splicing could be because of less loss due to NMD - or because they tend to be less important functionally, so the evolutionary pressure to ensure accurate splicing is weaker. One approach would be to compare the presence of cryptic splice sites in the sequence to actual molecular evidence of mis-splicing. If the 3-nt periodicity is simply a

result of NMD, then cryptic splice sites should be just as common at other sequence positions, they just won't show up in RNA-seq. If it is an evolved property, then the actual cryptic splice site sequences should also preferentially appear with the 3-nt periodicity.

5) Several comments regarding the regression models:

- a. In many cases, we see very small coefficients but significant P-values. As the sample sizes are very large here, even very small biologically meaningless effects may easily reach significance. It would therefore be better to rely not just on the P-value as a threshold but also on some sort of a second threshold based on the effect size. Predictors would have to reach both thresholds before they can leave "the grey zone" in the heat map.
- b. For both the regression model of the mis-splicing rate, as well as the model of TPM, given the type of data, it seems unlikely that normal linear regression would be reliable. For the mis-splicing rate, perhaps something like beta regression or quasibinomial regression would work better? For the TPMs, perhaps it's best to stick with normal linear regression but to work with the logarithm of the TPMs rather than the raw values.
- c. Methods should say whether or not the predictors were scaled (converted to Z-scores) prior to the model fitting. The plots in 4d and 5a-b are only meaningful if scaling was performed as the coefficients are not comparable otherwise. The same goes for the comparison between the coefficients of phastCONS vs CDTs scores in the main text.
- d. Did authors analyzed 42 tissue samples from a selection of 14K people? In that case, shouldn't tissue type and/or individual have random effects in the model?

Additional points that should be revised:

- 1) Page 3: whether or not splicing happens in the order of transcription, and whether it is constitutive or alternative, are issues that are related but not identical. The introduction makes it sound a bit like they are merely two sides of the same coin.
- 2) Page 5: is it meant that 25 intronic base pairs need to be recognized i.e. that this is the minimum intron length? Or does this number also include a portion of the flanking exons? It would be good to formulate this more explicitly.
- 3) Page 5: "Interestingly, we noted that the highest re-classification rates were observed amongst human brain tissues, on average $0.8 \pm 0.12\%$." Could this estimate be a typo, with the true value around 1.23%?
- 4) Page 5 (and elsewhere): what counts as alternative splicing and what counts as erroneous splicing is a philosophical matter. The only way to solve it would be through an evolutionary analysis to see which isoforms are conserved by natural selection. Just because a splice variant appears consistently and ends up being annotated does not mean that it is a functional alternative isoform. It may merely represent an error that is easy to make and that thus ends up made over and over again. Conversely, a lack of annotation does not necessarily imply non-functionality. The authors should express this point more clearly as the current formulation makes the distinction sound less arbitrary than it really is.
- 5) In accordance with the previous comment, the term "contamination rate" that is used sporadically through the text may be too much of a value statement.
- 6) For all P-values, the statistical test used should be reported in the main text, as well as whether the test was one- or two-tailed. For ranges of parameter estimates, the type of range should be reported (e.g. 95% confidence interval).
- 7) The abbreviation "CDTS" is not defined at first occurrence. It would also be good to provide a bit more detail as to how this metric is calculated.
- 8) Page 14: it feels unintuitive to convert this ratio to a percentage. It would be better to define the denominator as $j + y$, rather than just y .
- 9) Splicing patterns evolve rapidly. It may not be meaningful to work with a very deep alignment like phastCONS 20 as it is likely that the exon-intron architectures are rarely conserved that deep.

Reviewer #2

(Remarks to the Author)

Authors presented splicing inaccuracies (5'donors and 3'acceptors) of human protein-coding genes increase over aging. They found many novel 5'donors and 3' acceptors not reported in two GENCODE annotations, which display lower sequence conservation and lower read-supporting. They claimed that the inaccuracies may be caused by the decrease of RBPs. This study is the first report to systematically investigate the mis-splicing events in transcriptomes at the human population scale. Overall, the study is interesting, however there are some missing logics to claim the relationship between mis-splicing and aging and their causal factors. In addition, this study requires some additional analysis and experiments to deliver their conclusions as below.

(Major)

1. Authors analyzed ~14 control samples from 42 human body sites and collected reads partially mapped to the annotated transcripts (or annotated introns) and found 3,800K unique novel donor and acceptor junctions. They displayed lower sequence conservations and much lower supporting reads (~2 reads). The novel donor and acceptors displayed similar sequences with original ones. The highly expressed genes have more erroneous splicing. All results could indicate that the novel splice donor and acceptors could result from the sequencing errors and mis-alignments.

In fact, short-reads spanning introns can be aligned through anchored regions (splitted). If the anchor size is small, they tend to cause the mis-alignment, making mis-splicing informations.

They should use at least two independent reads supporting novel splice donor and acceptors to exclude results by the sequencing errors and mis-alignment. In addition, they have to test if the anchor size affects the results when they map the

reads to the references.

2. Related to 1, Authors claimed the majority of novel splice donors and acceptors are errors by showing the results that a very few novel splice sites were included in the newer GENCODE annotations. However, this is an indirect analysis and evidence. They should use more direct analysis and evidence for this. Maybe, they have to validate some randomly selected novel splice sites using experiments.

3. Authors used MaxEntScan scores to show the similarity of annotated and novel splice sites. However, they may want to show the sequence similarity between the annotated ones(original) and the novel sites, which could explain the mis-splicing by sequence-similarity directly. In addition, they may want to show the results (comparison of annotated vs novel) using SpliceAI scores.

4. Authors claimed that the mis-splicing increases over aging by reduction of RBPs by showing that the KD of splicing factors increases the mis-splicing and the splicing factors decrease over aging. However, this seems to make simple interpretation and some logically missing links. There are many different possible stories to explain how the RBPs can increase or decrease mis-splicing.

5. Related to 4, the isoforms with non-modulo3 value were ~62% of all mis-spliced isoforms. It could result from the exp. changes of NMD factors over aging, which is not surprising. They have to scrutinize how each RBP can contribute the mis-splicing.

6. Although this study first demonstrate the comprehensive view of mis-splicing in the population-level, there were some studies demonstrating age-/longevity-related mis-splicing in mice, fly, and *C.elegans*. They should cite them as well.

Reviewer #3

(Remarks to the Author)

The manuscript of García-Ruiz et al. reported on how splicing inaccuracies occur at different rates across introns and tissues and how they are primarily affected by the abundance of core components of the spliceosome and its regulators. They importantly found that splicing fidelity declines with age, resulting in a broad surge in intron retention and diverse spurious exon-exon junctions, which has not been thoroughly addressed. Many findings are consistent with previous studies on splicing timing and efficiency, highlighting that splicing is regulated in a cell-type-specific manner.

Overall, the manuscript is concise and well-written, providing innovative insights into splicing inaccuracies across various biological contexts based on thoroughly filtered data of high quality that support the stated conclusions. However, I do not have the expertise to evaluate the statistical and bioinformatic analyses performed. The findings of this study contribute to a better understanding of the regulation of splicing in different tissues, age groups and introns, and they contribute to the advancement of the field with potential clinical implications. I think it is a manuscript that fits well into the scope of Nature Communications.

The authors should consider some minor points:

1. As mentioned in the manuscript, NMD is crucial in ridding the transcriptome of aberrantly spliced transcripts, and its activity masks the impact of the actual mis-splicing landscape. I suggest that the authors incorporate the impact of NMD on mis-splicing by analyzing transcriptome samples without NMD factors (shRNA treatment of core NMD factors such as SMG1, UPF1, SMG6 and SMG7), similar to Fig. 5 of <https://doi.org/10.1186/s13059-021-02439-3> and Fig. 5 of <https://doi.org/10.1186/s13059-021-02309-y>. Such an analysis would address more clearly the impact of mis-splicing when NMD is compromised and would significantly enrich the study.

2. Along the same lines, an essential conclusion of the study is the correlation between ageing and the availability of splicing-related factors. In this view, it is recommended to address whether mis-splicing is related to the abundance of NMD factors at different ages, as this would mean that the identification of mis-spliced isoforms is not attributed only to splicing deficiency but also to the reduced availability of NMD factors.

3. The authors applied stringent filtering of reads to avoid including false-positive reads of splicing noise as described in the extended data Fig. 1. I would suggest including a short description of these criteria in the first paragraph of the results where authors refer to the extended data fig.1, as they contribute to the robustness of the study.

4. In an effort to implicate NMD, the authors present in Fig.5 factors that are erroneously stated as NMD factors (ribosomal proteins and general translation factors). This panel should be removed as these factors generally impact the transcriptome mostly in an NMD-independent manner.

5. There are reports of short-distance tandem sites giving rise to coding proteins that contribute to the protein evolution that the authors may consider discussing. In such cases, one should be cautious before characterizing such events as mis-splicing. Some examples are the following:
<https://doi.org/10.1371/journal.pbio.1001229>
<https://doi.org/10.1371/journal.pone.0096557>
<https://doi.org/10.1371/journal.pcbi.1008329>

Consider adding this notion to the discussion.

6. On p.6 (last two lines), the sentence "High sequence similarity between novel and annotated splice sites might be expected if these sites were near each other" was a bit unclear at first read. I suggest replacing the word "near" with "in close proximity".

7. Consider elaborating further on gene-level properties associated with mis-splicing (2nd paragraph of p.8), based on Fig.5.

8. On Fig.4a the y axis description should be positioned lower since it refers to both panels.

Version 1:

Reviewer comments:

Reviewer #1

(Remarks to the Author)

The authors have effectively addressed the primary technical and methodological points I raised, resulting in a significantly improved manuscript. While I have no reservations regarding the data quality, I maintain my view that the study's contribution to advancing the field is limited, and it lacks compelling evidence for biological relevance.

The authors emphasize the novelty of the concept of "splicing accuracy." However, it remains ambiguous how this concept differs from "mis-splicing," a phenomenon extensively documented in association with aging (see PMID: 36040386) and neurodegenerative diseases such as Alzheimer's disease (see PMID: 37336982).

(Remarks on code availability)

Reviewer #2

(Remarks to the Author)

I appreciate the authors' efforts to address and improve the manuscript based on the comments. They carefully and sincerely replied to the comments along with extensive additional analyses. However, the increase of mis-spliced transcripts over aging and diseases is significantly associated with the abundance of NMD factors, which are already documented multiple times before. It is still unclear how much or which splicing factors (RBPs) contribute the mis-splicing over the conditions. Without controlling NMD factor changes or controlled experimental validation, it is difficult to argue how much or which splicing factors (RBPs) impact the mis-splicing across age and disease.

(Remarks on code availability)

Reviewer #3

(Remarks to the Author)

After an intense peer review of an already interesting manuscript, the revised manuscript of García-Ruiz et al. has been considerably improved, and I support its publication in Nature Communications.

Addressing comments 1 and 2, the authors performed additional analysis and reported important evidence concerning the relationship between mis-splicing and NMD. Even though it is not the main focus of the manuscript, this is valuable evidence related to the broad community of genetic diseases. Therefore, I suggest highlighting their findings in the discussion and, if possible, illustrating this aspect in the Abstract and the Introduction.

More specifically, I suggest removing the phrase from the discussion and mentioning that authors do not consider NMD in their analysis (Fifthly, our analyses have not modelled the direct impact of NMD on mis-splicing activity). Instead, in light of their new analysis, I suggest pointing out that knocking down components of the NMD pathway led to a modest increase in the levels of acceptor splice-site noise and that ageing may be related to variation in NMD factors expression levels.

Previous evidence pointing in a similar direction is the following:

<https://doi.org/10.1186/s13059-021-02309-y>

<https://doi.org/10.1186/s13059-021-02439-3>

<https://doi.org/10.1371/journal.pgen.0020045>

I would like to commend the interesting work and review of the authors.

(Remarks on code availability)

Version 2:

Reviewer comments:

Reviewer #2

(Remarks to the Author)

Thank you for the repeated explanation and additional analysis for claiming that splicing-related RBPs give more impact on splicing inaccuracy over aging while NMD factors also contribute somewhat. However, I am still concerned about which RBPs are actually involved in the splicing accuracy of which genes. If all decreased RBPs contribute to splicing inaccuracy, then how can the mis-splicing of a particular gene be targeted to treat disease?

On the other hand, the authors' claim may be strengthened if they can analyze how much the non-NMD targets are affected by aging or by the decrease of splicing-related RBPs. To collect the non-NMD targets, they can consider publicly available RNA-seq data followed by UPF1 KD or KO.

(Remarks on code availability)

Reviewers' Comments to the Author

We thank the three reviewers for their helpful and detailed comments. We have carefully considered all suggestions (see below) with the original comments in black and our detailed responses highlighted in blue.

Referee reports

Reviewer #1 (Remarks to the Author):

This manuscript reports a careful computational analysis of RNA splicing across human tissues, relying merely on RNA-seq data that already exist in public databases. The results are sound and the figures are creative and well-thought-through. The manuscript is well-written and structured. However, the study does not contribute a significant advance to the splicing field. It has long been established that alternative splicing (which includes most of the reported "mis-spliced" events, except for cryptic splicing caused by DNA mutations) is tissue-specific, and this involves the differential expression/activity of splicing regulatory proteins. Additionally, a major caveat of this study concerns the lack of evidence for biological relevance of the findings. Indeed, many of the "mis-spliced" transcripts represent a minor proportion compared to normally processed transcripts. Thus, they may result from splicing noise and have no physiological consequence either for ageing or for disease.

We thank the reviewer for recognising that our results are sound, that the figures are creative and that the manuscript is well-organised and written. Indeed, the reviewer is correct that the findings are based on publicly available RNA-seq data, but we wish to highlight that this involves data on over 17,000 samples including those generated from GTEx, ENCODE, and a further separate data set now incorporated into the revised manuscript (Gene Expression Omnibus: GSE89482). However, we respectfully disagree with the reviewer's assessment that our study does not i) contribute a significant advance to the field, and ii) that the biological relevance of the work is limited.

Focusing on the first point, the reviewer is of course correct that it has long been known that alternative splicing is tissue-specific and that the activity of splicing regulators is key to this process. However, this manuscript is focused on the question of whether the **accuracy** of splicing varies across tissues or tissue states (e.g. ageing), and this need not be a function of the expression of splicing regulatory proteins. In fact, as the reviewer himself raises, the conventional wisdom has been that such errors in splicing occur as a result of "DNA mutations" and most of the work has focused on high-effect variants within the context of rare diseases. Furthermore, others in the field, notably the fellow reviewers of this manuscript, have suggested that variability in splicing accuracy across tissues could be a function of the activity of the nonsense-mediated decay pathway. Consequently, even after sampling opinions from the reviewers of this manuscript, the fact that we find that errors in the accuracy of the splicing machinery arise from differences in the expression of splicing regulatory proteins expressed at physiological levels appears to be far from established and therefore this is an advance for the field.

Focusing on the reviewer's second concern, namely the biological relevance of splicing errors, we again disagree that the low abundance of such events inevitably means that they are unlikely to have any physiological consequence. We base our view, firstly, on the fact that irrespective of abundance,

robust patterns in mis-splicing events have the potential to reveal important biology. For example, our analyses demonstrate that contrary to expectation (including that of that reviewer), splicing noise is not primarily driven by DNA mutations, and that such noise is consistently more frequent at 3' acceptor splice sites. In fact, we note that in the context of ageing, there is growing interest in the impact of globally subtle but detectable changes in multiple aspects of transcriptional processing, including the relationship between transcriptional rate increases with age and splicing fidelity (Debès, C., Papadakis, A., Grönke, S. et al. 2023). Secondly, and perhaps more importantly, we find that splicing errors increase with age in a heterogeneous manner across introns with a disproportionate impact in the human brain on genes involved in synaptic function. Therefore, while the effect of age on splicing may appear low globally, the effect on key processes may be significant.

In order to explore this point further in a disease context, we have now extended our analyses to include an additional RNA-sequencing data set of 98 Alzheimer's disease patients and neurologically normal (control) post-mortem human brain samples (BA Friedman et al., 2018). Using similar analytical approaches (see Methods "*Alzheimer's disease/control short-read RNA-sequencing data download and processing*", Results "*Splicing integrity decreases in genes enriched for synaptic functions in Alzheimer's Disease*") to those used for ageing analyses, we found significant increases in the levels of mis-splicing activity across the transcriptome in brain samples from AD patients as compared to controls (Fig.8a,b) at both donor (one-tailed paired Wilcoxon signed rank test, eff.size=0.052, $P<0.001$) and acceptor sites (one-tailed paired Wilcoxon signed rank test, eff.size=0.054, $P<0.001$). These mis-splicing events were more likely to be capable of generating frame-shift events, indicating a deleterious effect (Fig.8c). We observed that the genes containing higher frequencies of splicing inaccuracies in AD samples were enriched for synaptic functions (Fig.8d,e). Given that cognitive impairment in Alzheimer's disease is thought to precede neuronal loss and is driven by synaptic dysfunction, this finding implies that splicing integrity for key synaptic genes is a component of the pathophysiology of disease in AD. We have included these results in the amended manuscript (see Methods "*Alzheimer's disease/control short-read RNA-sequencing data download and processing*", Results "*Splicing integrity decreases in genes enriched for synaptic functions in Alzheimer's Disease*" and main Fig.8).

Overall, we believe that these results provide additional evidence for the biological relevance of the analyses, techniques and results provided in this manuscript, and are of broad interest.

MAJOR COMMENTS:

- 1) It would be good to see an analysis not just of the prevalence of mis-splicing events in different samples but also of the extent to which the specific events are shared between samples. It would help assess to what extent the variability in mis-splicing patterns between samples is quantitative vs qualitative.

We thank the reviewer for this valuable comment, which we have implemented. Changes can be found in the amended manuscript, within the Results section ("*Novel donor and acceptor junctions are commonly detected and exceed the number of unique annotated introns by an average of 11-fold*") Supplementary Figures section (Supplementary Figures 2 and 3). In Supplementary Figure 2, we display the extent to which each specific novel junction is shared across all samples of each GTEx body site. Novel donor and novel acceptor junctions have been considered collectively. As expected, given that novel junctions are largely capturing splicing noise, the majority of novel junctions are unique to an individual or are shared across a very low number of individuals across all tissues. We note that a similar analysis of annotated junctions (Supplementary Figure 3)

demonstrates that the vast majority of these annotated events are shared across a high number of individuals in all tissues.

- 2) When running MaxEntScan scores, it's unclear whether the authors used the genomic sequence of the reference genome or the one inferred from the genomic data of GTEx. In case of the former, authors should note that the reference genome for those cases is not representative of the biologic conditions where that mis-splicing was generated. Splice sites can be straightened or weakened by the presence of variants that are not currently being considered.

We thank the reviewer for this comment and agree that it is important to highlight to the reader that the calculation of MES scores is based on the reference hg38. Hence, we have amended the manuscript accordingly (section Online Methods, "*MaxEntScan score analyses*"). Unfortunately, even using genomic data from GTEx, will not be representative of the genomic sequence where mis-splicing was generated since this would require genomic data from each tissue rather than a single genomic sequence per individual, which is what is available.

We agree with the reviewer that using a reference genome to infer the strength of the splicing motif sequences studied cannot capture the existence of potential germline or somatic nuclear variation at an individual level. However, we think it is helpful to point out that we considered the potential contribution of germline genetic variation in our analyses. For this reason, we included a measure of DNA sequence constraint in humans, termed the mean context-dependent tolerance score (CDTS) (J di Iulio et al., 2018), as a covariate within the mis-splicing regression models to evaluate whether germline genetic variation at exon-intron boundaries could influence the levels of mis-splicing. Furthermore, we note that in the original version of this manuscript, we investigated the contribution of somatic variation on the frequency of mis-splicing using data from skin exposed to sunlight tissue and unexposed skin, with the former known to have a higher somatic mutation load as a result of UV light exposure (K Yizhak et al., 2019). We did not find any significant differences in mis-splicing rates across these two tissue types, suggesting that somatic variation is not a major driver of mis-splicing (see Results section "*Accuracy in splicing is affected by RNA-binding protein (RBP) expression changes*").

- 3) The authors state that "sequence conservation in genomic regions flanking the 5' and 3'ss had the largest effect on splicing accuracy". This cannot be true, at least in a mechanistic sense. (Note that, in accordance with Figure 1b, I am interpreting this statement to be not about how well a given splice site matches the splice site consensus but rather about interspecies conservation - in the figure, the human splice site sequences are identical for the two examples.) The splicing machinery does not know anything about what the sequence is in other species. The conservation of the splice sites is merely an indicator that these introns are functionally important, and so it makes sense that there would be a pressure to splice them accurately. But the actual mechanistic determinants of why the splicing is more or less accurate for a given intron must be elsewhere – in cis-regulatory elements, in chromatin structure, in the exon-intron architecture...

We thank the reviewer for raising this important point. We could not agree with the reviewer more that the level of inter-species sequence conservation at genomic regions flanking 3' and 5' sites cannot directly impact splicing fidelity and is likely to be an indicator of the functional importance of the introns. This makes it only more surprising that sequence conservation around splice sites should have a highly variable effect across human tissues on mis-splicing rates, and this is the reason for our focus on this term (Fig.5a,b). In fact, like the reviewer, we reasoned that the sequence itself (which

obviously does not change across tissues) is insufficient to explain the observed variability. Rather, we reasoned that the variability was generated elsewhere and could be due to tissue-specific expression levels of RBPs involved in splicing regulation. We have amended the text to make this point clearer to the reader (Results section: *“Accuracy in splicing is affected by RNA-binding protein (RBP) expression changes”*).

We also specifically tested this hypothesis by evaluating the effect of changes in the expression levels of 54 splicing-related, spliceosomal and exon-junction genes on mis-splicing ratios using the data provided by ENCODE on the sequencing of cell lines following gene knockdown (Results section of the manuscript *“Accuracy in splicing is affected by RNA-binding protein (RBP) expression changes”*, and Methods section *“Analysing shRNA knockdown of RBPs followed by RNA-sequencing data from ENCODE”*). Results from this analysis indicated that accuracy in splicing could indeed be affected by RNA-binding protein (RBP) expression and that differences in splicing accuracy across tissues could be explained in this way. To add additional support to the hypothesis that RBP expression differences play a major role in transcriptome-wide splicing accuracy, we have now run additional analyses using RBP-RNA interactions defined using CLIP-seq data. This analysis aimed to determine whether the annotated introns with the highest levels of mis-splicing in a given knockdown experiment, also contained a higher number of binding sites for that RBP as compared to other annotated introns. This analysis was performed by linking ENCODE gene knockdown data with CLIP mRNA binding data from the ENCORI⁴ platform, as both sources of data had been created from K562 and HepG2 cell lines. Using this approach we were able to study 15 of the 54 genes investigated by ENCODE. We found that annotated introns with increased mis-splicing under knockdown conditions had a significantly higher density of binding sites for that gene product, either in close proximity (+/-100 bp) or within their intronic sequence, as compared to other annotated introns (Fig.6c, Supplementary Table 6). This result suggests that variability in the expression levels of splicing-related and spliceosomal genes alters splicing accuracy and could explain tissue differences in splicing noise. We have included these results in the revised manuscript in sections (Results: *“Accuracy in splicing is affected by RNA-binding protein (RBP) expression changes”*, Methods *“Analysing shRNA knockdown of RBPs followed by RNA-sequencing data from ENCODE”* and *“Analysing RBP-RNA interactions of binding sites derived from CLIP-seq data”*, main Fig.6c and Supplementary Table 6).

- 4) Both for the phastCONS as well as the CDTs analysis, the authors use very short sequence regions that would not necessarily contain all the relevant cis-regulatory information. It would be interesting to include conservation/polymorphism levels within these larger regions as predictors in the regression models to see if perhaps the coefficient for splice site conservation decreases.

We agree with the reviewer that sequences at a distance from a given intron are likely to be involved in regulating its splicing and consequently the genomic window used to calculate phastCONS and CDTs scores could be widened. However, we note that i) there is no clear consensus on the distribution of cis-acting splicing elements around introns and ii) increasing the window size for the calculation of the phastCons and CDTs scores will result in a smaller set of introns being studied. The latter is because introns that are shorter than double the window size established would need to be discarded to avoid studying overlapping sequences when modelling mis-splicing activity at both splice sites. To balance these two concerns, we replaced the 35 bp window size originally used in the first version of this manuscript and replaced it with a 100 bp, as proposed by Wainberg and colleagues (M Wainberg et al., 2016), who posit that regulatory sequences are predominantly within 100 bp of the splice sites. Consequently, only introns larger than 200 bp were included in the major

analyses, resulting in the discard of 31,670 annotated introns. While these changes made no substantive difference to the original results obtained for the phastCons scores, we observed an increase in the significance of the association between the context-dependent tolerance (CDTS) scores and mis-splicing rates. Specifically, we found that CDTS scores positively affected mis-splicing rates at the 5'ss (MSRD=1.01 [1.01,1.01]; MSRA=1.01 [1.01,1.02]) and 3'ss (MSRD=1 [1,1.01]; MSRA=1.01 [1.01,1.01]). Given that positive CDTS values indicate poorly constrained sequences amongst humans, these results suggest that high sequence fidelity in the 100 bp intronic sequence neighbouring exon-intron junctions is required to maintain splicing accuracy.

We have amended the manuscript accordingly, and changes corresponding to this comment can be found in the Methods section: *“Using Zero-Inflated Poisson Regression models to predict the rate of mis-splicing activity”*, Results section *“High sequence fidelity in the vicinity of exon-intron junctions is required to maintain splicing accuracy”*, main Fig 4.d, main Fig 5.a,b.

- 5) A critical issue is how much mis-splicing is missed because of NMD. The authors do two analyses that partially address this point but both also have alternative interpretations. Firstly, the 3-nucleotide periodicity of the novel splice junctions in coding regions could reflect that the other mis-splicing events get gnawed up by NMD – **or it could reflect some property of the genetic code, perhaps even some sort of evolved robustness against splicing errors**. Secondly, the fact that non-coding genes show more mis-splicing could be because of less loss due to NMD – or because they tend to be less important functionally, so the evolutionary pressure to ensure accurate splicing is weaker. One approach would be to compare the presence of cryptic splice sites in the sequence to actual molecular evidence of mis-splicing. If the 3-nt periodicity is simply a result of NMD, then cryptic splice sites should be just as common at other sequence positions, they just won't show up in RNA-seq. If it is an evolved property, then the actual cryptic splice site sequences should also preferentially appear with the 3-nt periodicity.

We thank the reviewer for this very useful comment and agree that our analyses only partially address the contribution of NMD. The reviewer specifically suggests that we identify cryptic splice sites. However, to the best of our knowledge, there is no publicly available dataset providing such information in a genome-wide manner. Nevertheless, given that it is known that cryptic splice sites have a higher degree of motif sequence similarity to annotated splice sites as compared to non-cryptic sequences (Roca et al. 2003), we found that the most feasible way of following the reviewer's suggestion was through a reassessment of MaxEntScan scores (MES). Hence, we obtained the delta MES of the novel splice sites located at distances divisible by three from their annotated pairs and compared them to scores generated from all other novel junctions. Focusing on novel junctions that were located up to 100 bp from their annotated pairs, we found that the novel junctions located at positions which were multiples of three ($\text{mod}3=0$) did not have a significantly higher degree of motif sequence similarity to annotated sites, as compared to all other novel junctions (i.e. $\text{mod}3=1$, $\text{mod}3=2$) (one-tailed Wilcoxon Rank-sum test, $P=1$) (Supplementary Figure 9). This result suggests that the higher densities of novel acceptor splicing events detected every 3 bp from annotated splice sites are not primarily produced by a property of the genetic code.

We have amended the manuscript accordingly to include this analysis, and changes corresponding to this comment can be found in the Results section *“Novel junctions associated with protein-coding transcripts are predicted to be deleterious in 63.5% of cases”*, and Supplementary Figure 9.

- 6) Several comments regarding the regression models:

- a. In many cases, we see very small coefficients but significant P-values. As the sample sizes are very large here, even very small biologically meaningless effects may easily reach significance. It would therefore be better to rely not just on the P-value as a threshold but also on some sort of a second threshold based on the effect size. Predictors would have to reach both thresholds before they can leave “the grey zone” in the heat map.

We thank the reviewer for this useful comment. Since even small effect sizes can provide meaningful biological insights, we respectfully disagree with the application of a second threshold.

- b. For both the regression model of the mis-splicing rate, as well as the model of TPM, given the type of data, it seems unlikely that normal linear regression would be reliable. For the mis-splicing rate, perhaps something like beta regression or quasibinomial regression would work better?

We thank the reviewer for this useful and valuable comment. We considered the reviewer’s suggestion of using beta and quasibinomial regression. However, we could not apply beta regression because this method is not well-suited for modelling zeros, and the independent variable used in the models, the MSR_D and MSR_A , are half-bounded measures between [0,1). We also tried to use quasibinomial regression, but the zero mass of the data was likely to bias the modelling.

Consequently, and given the characteristics of the MSR measures, we concluded that the most suitable regression model to apply in this case would be Zero-Inflated Poisson (ZIP) regression. One of the characteristics of the ZIP method is that it produces two different models: a count model to predict the count data along with some of the zeros, and a logit model to predict the excess of zeros. While we think that ZIP is most appropriate, we found that some dependent variables could no longer be studied due to correlations among them, these included “*gene length*” and “*gene TPM*”. These changes have been included in the amended manuscript (Methods section “*Using Zero-Inflated Poisson Regression models to predict the rate of mis-splicing activity*”, Results section: “*High sequence fidelity in the intronic sequences neighbouring exon-intron junctions is required to maintain splicing accuracy*”, and corresponding main figures Fig4.d and Fig5.a,b).

- c. For the TPMs, perhaps it’s best to stick with normal linear regression but to work with the logarithm of the TPMs rather than the raw values.

We thank the reviewer for this suggestion. We agree with the reviewer that TPM values should be transformed to the log scale before model fitting, and we have applied this suggestion accordingly in the relevant sections. Changes can be found in the amended manuscript (Results section “*Increasing age is associated with increasing levels of mis-splicing*”, and Methods section “*Changes in RBP expression levels with age in brain tissue*” and Supplementary Tables 7 & 8).

- d. Methods should say whether or not the predictors were scaled (converted to Z-scores) prior to the model fitting. The plots in 4d and 5a-b are only meaningful if scaling was performed as the coefficients are not comparable otherwise. The same goes for the comparison between the coefficients of phastCONS vs CDTS scores in the main text.

We thank the reviewer for this very helpful comment. Focusing on the reviewer’s comment about Fig.4d, we have not scaled the predictors prior to model fitting and this is now clarified in the text. The reason for this is that we think that scaled values would be harder for a reader to interpret. However, we agree that this means that predictors cannot be meaningfully compared within this

plot. We have amended the text accordingly (Methods section *“Using Zero-Inflated Poisson Regression models to predict the rate of mis-splicing activity”*, and Results section *“High sequence fidelity in the vicinity of exon-intron junctions is required to maintain splicing accuracy”*).

Focusing on the reviewer’s comment about Fig.5a,b, we are unable to use scaled values in this context. This is because the coefficient values produced relate to tissue-specific regression models which have tissue-specific distributions. However, the purpose of these figures was not to compare coefficients of different predictors, but rather coefficients of the same predictor across tissues. We have amended the text to clarify this point (Results sections *“High sequence fidelity in the vicinity of exon-intron junctions is required to maintain splicing accuracy”* and *“Accuracy in splicing is affected by RNA-binding protein expression changes”*).

- e. Did authors analyzed 42 tissue samples from a selection of 14K people? In that case, shouldn’t tissue type and/or individual have random effects in the model?

We thank the reviewer for this comment. We think the reviewer may have confused some of the metrics. In this project, we analysed the splicing activity of 332,571 annotated introns found across ~14K human samples from 900 unrelated individuals and 40 tissues. We have analysed each tissue independently rather than collectively, and there are considerable differences in the numbers of individuals contributing to each tissue data set. Consequently, modelling the random effects of individuals would not be possible.

Additional points that should be revised:

- 1) Page 3: whether or not splicing happens in the order of transcription, and whether it is constitutive or alternative, are issues that are related but not identical. The introduction makes it sound a bit like they are merely two sides of the same coin.

We thank the reviewer for clarifying this and have amended the text accordingly (Introduction section).

- 2) Page 5: is it meant that 25 intronic base pairs need to be recognized i.e. that this is the minimum intron length? Or does this number also include a portion of the flanking exons? It would be good to formulate this more explicitly.

We thank the reviewer for this point of clarification. Regarding 25 intronic base pairs, we are referring to the minimum intron length (this does not include flanking exons). We have clarified this (Introduction section, Methods section *“GTEx v8 RNA-sequencing data download and processing”*).

- 3) Page 5: “Interestingly, we noted that the highest re-classification rates were observed amongst human brain tissues, on average $0.8 \pm 0.12\%$.” Could this estimate be a typo, with the true value around 1.23%?

We agree with the reviewer that this is a typographical error and have corrected the text accordingly (Results section *“Over 98% of novel donor and acceptor junctions are likely to be generated through splicing errors”*).

- 4) Page 5 (and elsewhere): what counts as alternative splicing and what counts as erroneous splicing is a philosophical matter. The only way to solve it would be through an evolutionary analysis to see which isoforms are conserved by natural selection. Just because a splice variant appears consistently and ends up being annotated does not mean that it is a functional alternative isoform. It may merely represent an error that is easy to make and that thus ends up made over and over again. Conversely, a lack of annotation does not necessarily

imply non-functionality. The authors should express this point more clearly as the current formulation makes the distinction sound less arbitrary than it really is.

We thank the reviewer for this very helpful comment. We could not agree more with the reviewer that a splice event being part/not part of annotation does not necessarily mean correct/incorrect splicing activity. To make this point clearer to the reader, we have amended the text and changes can be found in the *Discussion* section.

- 5) In accordance with the previous comment, the term “contamination rate” that is used sporadically through the text may be too much of a value statement.

We thank the reviewer for this very helpful comment and agree. We have replaced the term “contamination rate” with “re-classification rates”. Text changes and amended figures can be found in the Results section “*Over 98% of novel donor and acceptor junctions are likely to be generated through splicing errors*” and “*Calculating the re-classification rates across multiple versions of the Ensembl transcriptome*”, Fig.2a and Supplementary Fig. 5.

- 6) For all P-values, the statistical test used should be reported in the main text, as well as whether the test was one- or two-tailed. For ranges of parameter estimates, the type of range should be reported (e.g. 95% confidence interval).

We thank the reviewer for this comment. We have amended the manuscript accordingly and indicated the statistical test used and if it was one- or two-tailed. For ranges of parameter estimates, we have included the range of confidence intervals when using regression analysis. Changes can be found throughout the text and in **Fig. 4d**.

- 7) The abbreviation “CDTS” is not defined at first occurrence. It would also be good to provide a bit more detail as to how this metric is calculated.

We thank the reviewer for this suggestion which we have implemented. We have defined the abbreviation CDTS as soon as it appears (Results section “*High sequence fidelity in the intronic sequences neighbouring exon-intron junctions is required to maintain splicing accuracy*”), and added further details about this score within the Methods section “*Using Zero-Inflated Poisson Regression models to predict the rate of mis-splicing activity*”.

- 8) Page 14: it feels unintuitive to convert this ratio to a percentage. It would be better to define the denominator as $j + y$, rather than just y .

We thank the reviewer for this suggestion which we have implemented. We have amended the manuscript accordingly (Methods section “*Calculating the re-classification rates across multiple versions of the Ensembl transcriptome*” and Results section “*Over 98% of novel donor and acceptor junctions are likely to be generated through splicing errors*”, and figures Fig.2a and Supplementary Fig. 5.

- 9) Splicing patterns evolve rapidly. It may not be meaningful to work with a very deep alignment like phastCONS 20 as it is likely that the exon-intron architectures are rarely conserved that deep.

We again thank the reviewer for this valuable comment, which we have implemented. We have replaced phastCons20 conservation scores and used phastCons17 instead. While phastCons20 scores were calculated using multiple alignments of 19 genomes (16 primates, Tree Shrew, Mouse and Dog) to the human genome, the phastCons17 scores contained conservation scores only for 16 primate genomes to the human genome, so reducing the evolutionary distance among the genomes used

(<https://hgdownload.soe.ucsc.edu/goldenPath/hg38/phastCons17way/>). Despite this change, using phastCons17 scores made no substantive difference to the results. Changes can be found in the amended manuscript (Results section *"High sequence fidelity in the vicinity of exon-intron junctions is required to maintain splicing accuracy"*, Methods section *"Using Zero-Inflated Poisson Regression models to predict the rate of mis-splicing activity"* and figures, Fig.4d, Fig.5a-b).

Reviewer #2 (Remarks to the Author):

Authors presented splicing inaccuracies (5'donors and 3'acceptors) of human protein-coding genes increase over aging. They found many novel 5'donors and 3' acceptors not reported in two GENCODE annotations, which display lower sequence conservation and lower read-supporting. They claimed that the inaccuracies may be caused by the decrease of RBPs. This study is the first report to systematically investigate the mis-splicing events in transcriptomes at the human population scale. Overall, the study is interesting, however there are some missing logics to claim the relationship between mis-splicing and aging and their casual factors. In addition, this study requires some additional analysis and experiments to deliver their conclusions as below.

We thank the reviewer for his/her insightful comments and for recognising that this is the first study to systematically investigate mis-splicing at a population scale. We agree that the original study would be made stronger after following his/her recommendations and, consequently, we have thoughtfully considered all of them.

(Major)

1. Authors analyzed ~14 control samples from 42 human body sites and collected reads partially mapped to the annotated transcripts (or annotated introns) and found 3,800K unique novel donor and acceptor junctions. They displayed lower sequence conservations and much lower supporting reads(~2 reads). The novel donor and acceptors displayed similar sequences with original ones. The highly expressed genes have more errorneous splicing. All results could indicate that the novel splice donor and acceptors could resulted from the sequencing errors and mis-alignments. In fact, short-reads spanning introns can be aligned through anchored regions(splitted). If the anchor size is small, they tend to cause the mis-alignment, making mis-splicing informations. They should use at least two independent reads supporting novel splice donor and acceptors to exclude results by the sequencing errors and mis-alignment. In addition, they have to test if the anchor size affects the results when they map the reads to the references.

We thank the reviewer for this really useful comment. We also assume the reviewer's comments contain an error in that we analysed ~14,000 samples from 40 human body sites (rather than 14 samples).

As suggested by the reviewer, we have now assessed mis-splicing activity by considering only novel donor and acceptor junctions supported by two independent split reads in at least two of the 14K samples of the 40 GTEx tissues studied. Applying this filter resulted in a significant reduction (73%) in the quantity of data available for study. More specifically, only 273,827 annotated introns, 543,476 novel donor and 705,474 novel acceptor junctions remained (in comparison to 324,956 annotated and 3,865,268 novel junctions reported in the original version of this manuscript). Importantly, despite this dramatic reduction in the quantity of available data, when we evaluated the distribution of mis-splicing activity around annotated sites we still observed the same patterns in splicing noise as those reported in the original version of this manuscript (see Supplementary Figure 6) In particular, mis-splicing rates remained higher at the acceptor than at donor splice sites, the distribution of mis-splicing events at these sites followed a similar distribution when considered collectively, and mis-splicing rates were lower amongst introns exclusively used by protein-coding transcripts. Consequently, we believe that discarding ~73% of the data available would represent an excessive

data loss that cannot be easily justified when the patterns of splicing noise originally reported in this manuscript remained in such a robust manner. Therefore, we have based our primary results on our original approach, but have reported the more conservative analysis within the amended manuscript (see Methods section “*GTEX v8 RNA-sequencing data download and processing*”, Results section “*Mis-splicing is more common at acceptor than donor splice sites*” and Supplementary Figure 6).

Focusing on the reviewer’s second suggestion, we evaluated the effect that the anchor size configured in the aligner used during RNA-seq data mapping could have on the results reported in mis-splicing activity. RNA-sequencing data from the GTEx v8 project was aligned using STAR (version 2.7.3a) with default parameters, in which the anchor size was set at a value of 5 (“--outSJfilterOverhangMin” = 5, <https://gensoft.pasteur.fr/docs/STAR/2.7.3a/STARmanual.pdf>). Since re-aligning the raw RNA-seq data from the whole GTEx v8 project would have resulted in a considerable increase in the time required to complete the evaluation and assessment of this suggestion, instead we re-assessed 216 shRNA control experiments followed by RNA-sequencing data downloaded from the ENCODE platform. We used a minimum anchor length of 8 bp to call the presence of a novel junction (RegTools, <https://regtools.readthedocs.io/en/latest/>), which represented an increase in stringency of 3 bp with respect to the anchor used in the GTEx v8 data. Using the re-aligned ENCODE shRNA data across 216 control shRNA knockdown experiments (see Methods sections “*Analysing shRNA knockdown of RBPs followed by RNA-sequencing data from ENCODE*”), we reproduced the analyses included in this manuscript and observed that the patterns of splicing noise robustly remained even after using a stricter anchor size of 8 bp (Supplementary Figure 7). Consequently, we concluded that the anchor size configured during the alignment of the GTEx v8 dataset was not a driver of the patterns of splicing noise reported in this manuscript. We have included this information within the revised manuscript (see Results section “*Mis-splicing is more common at acceptor than donor splice sites*” and Supplementary Figure 7).

2. Related to 1, Authors claimed the majority of novel splice donors and acceptors are errors by showing the results that a very few novel splice sites were included in the newer GENCODE annotations. However, this is an indirect analysis and evidence. They should use more direct analysis and evidence for this. Maybe, they have to validate some randomly selected novel splice sites using experiments.

We thank the reviewer for this comment. To add further support for the hypothesis that most of the low-expressed novel donor and acceptor junctions studied in this manuscript represent splicing noise likely produced by the personal transcriptome of each sample studied, we have run an additional analysis of a new long-read RNA-seq data set. We used targeted long-read RNA-sequencing data of *GBA1* across 12 human brain regions and multiple individuals (Gustavsson et al. 2022) generated on the Pacific Biosciences Sequel II platform, with RNA samples with RINs > 8 and stringent quality control for full-length transcript detection.

From the 2,368 unique *GBA1* novel transcripts originally reported by Gustavsson et al. 2022, we observed that 1,328 novel transcripts contained at least one novel donor or acceptor splicing event. Following the methods proposed by Gustavsson et. al (2023), we calculated the normalised expression of these 1,328 unique *GBA1* transcripts across all samples. We found that all but 15 of these novel transcripts had a normalised expression of less than 0.5 indicating that the vast majority of transcripts containing a novel donor or acceptor event accounted for less than 0.5% of all transcripts mapping to the locus.

While ideally, we would wish to expand this analysis beyond the *GBA1* locus and include all findings in the manuscript. Unfortunately, there are no publicly available genome-wide long-read

RNA-sequencing data sets with the necessary sequencing depth. Consequently, this analysis is reported here for the benefit of the reviewer but is not included in the revised manuscript.

3. Authors used MaxEntScan scores to show the similarity between annotated and novel splice sites. However, they may want to show the sequence similarity between the annotated ones(original) and the novel sites, which could explain the mis-splicing by sequence-similarity directly. In addition, they may want to show the results (comparison of annotated vs novel) using SpliceAI scores.

We thank the reviewer for this very useful comment. Focusing on the reviewer's first suggestion, given that the modal distance for detection of novel splice sites to their annotated pairs is approximately 4 bp across all GTEx tissues (see Results section "*Novel junctions associated with protein-coding transcripts are predicted to be deleterious in 63.5% of cases*", Supplementary Table 2), high levels of sequence similarity across a given genomic window between each junction pair will inevitably be high. Consequently, we think that the sequence similarity analysis suggested by the reviewer will be of limited value so we have not implemented it.

Focusing on the reviewer's second suggestion, we note that SpliceAI requires genotype calls and WES/WGS are protected data within GTEx that require a long approval process, so we have not been able to use SpliceAI in this context. Nonetheless, we believe that using MES is sufficient for the purposes of this analysis since the two methods provide largely concordant results with the majority of discrepancies in prediction between methods reported for variants located in exons alone (<https://doi.org/10.3390/genes12091308>).

4. Authors claimed that the mis-splicing increases over ageing by reduction of RBPs by showing that the KD of splicing factors increases the mis-splicing and the splicing factors decrease over aging. However, this seems to make simple interpretation and some logically missing links. There are many different possible stories to explain how the RBPs can increase or decrease mis-splicing.

We thank the reviewer for this very useful comment. The reviewer is correct that establishing a direct relationship between RBP activity and mis-splicing is important. Since this can be studied through CLIP data, we linked the ENCODE shRNA-sequencing experiments analysed in the original manuscript with RBP-mRNA interactions derived from CLIP-seq data from the ENCORI (JH Li et al. 2014) database (both types of data had been created from matching K562 and HepG2 cell lines). We aimed to evaluate whether the annotated introns with increasing levels of mis-splicing in samples which had undergone knockdown of a given RBP also contained higher binding site densities for that RBP as compared to annotated introns with stable splicing. Importantly, we found that the annotated introns with increased levels of mis-splicing under knockdown conditions also had higher RBP binding site densities (as calculated using a 100 bp window) supporting a direct relationship between RBP expression, activity and mis-splicing rates (Fig.6c, Supplementary Table 6). We have included these results in the amended manuscript in the Results section "*Accuracy in splicing is affected by RNA-binding protein (RBP) expression changes*", Methods section "*Analysing RBP-RNA interactions of binding sites derived from CLIP-seq data*", main Fig.6c, and Supplementary Table 6.

5. Related to 4, the isoforms with non-modulo3 value were ~62% of all mis-spliced isoforms. It could result from the exp. changes of NMD factors over aging, which is not surprising. They have to scrutinize how each RBP can contribute the mis-splicing.

We thank the reviewer for this insightful comment. We agree with the reviewer in that the manuscript would benefit from evaluating how age-related expression changes of NMD factors affect mis-splicing activity. We selected *UPF1* and *UPF2*, and the proteins SMG-1, SMG-6 and SMG-7 since the three latter are known to control *UPF1* phosphorylation. We next built a linear regression model for each of the aforementioned genes to predict the relationship between their log10 level of expression, as measured through their TPM values, across the samples of each GTEx tissue studied. As dependent variables, we selected a set of covariates providing information about the sample, which included age (see Methods section: “*Changes in RBP expression levels with age*”). Results from this analysis indicated that increasing “age” was significantly associated with reduced expression of multiple NMD genes (see Supplementary Figure 18). This suggests that the reviewer is correct that at least a subset of the mis-spliced transcripts observed with age across GTEx samples could be due to reduced degradation of mis-spliced transcripts. In addition, we followed the reviewer's suggestion and assessed how each individual RBP contributes to mis-splicing by linking ENCODE shRNA-sequencing knockdown data with RBP-mRNA interactions derived from CLIP-seq data from the ENCORI database (see response above).

6. Although this study first demonstrate the comprehensive view of mis-splicing in the population-level, there were some studies demonstrating age-/longevity-related mis-splicing in mice, fly, and *C.elegans*. They should cite them as well.

We thank the reviewer for this comment. We have amended the manuscript and addressed this comment in the *Introduction* section. We have cited different studies that have demonstrated age-longevity-related mis-splicing changes in the organisms proposed by the reviewer.

Reviewer #3 (Remarks to the Author):

The manuscript of García-Ruíz et al. reported on how splicing inaccuracies occur at different rates across introns and tissues and how they are primarily affected by the abundance of core components of the spliceosome and its regulators. They importantly found that splicing fidelity declines with age, resulting in a broad surge in intron retention and diverse spurious exon-exon junctions, which has not been thoroughly addressed. Many findings are consistent with previous studies on splicing timing and efficiency, highlighting that splicing is regulated in a cell-type-specific manner.

Overall, the manuscript is concise and well-written, providing innovative insights into splicing inaccuracies across various biological contexts based on thoroughly filtered data of high quality that support the stated conclusions. However, I do not have the expertise to evaluate the statistical and bioinformatic analyses performed. The findings of this study contribute to a better understanding of the regulation of splicing in different tissues, age groups and introns, and they contribute to the advancement of the field with potential clinical implications. I think it is a manuscript that fits well into the scope of Nature Communications.

We thank the reviewer for his/her enthusiasm and appreciation of the value of this study.

Comments for the Author

The authors should consider some minor points:

1. As mentioned in the manuscript, NMD is crucial in ridding the transcriptome of aberrantly spliced transcripts, and its activity masks the impact of the actual mis-splicing landscape. I suggest that the authors incorporate the impact of NMD on mis-splicing by analyzing transcriptome samples without NMD factors (shRNA treatment of core NMD factors such as SMG1, UPF1, SMG6 and SMG7), similar to Fig. 5 of <https://doi.org/10.1186/s13059-021-02439-3> and Fig. 5 of <https://doi.org/10.1186/s13059-021-02309-y>. Such an analysis would address more clearly the impact of mis-splicing when NMD is compromised and would significantly enrich the study.

We thank the reviewer for this helpful comment. Focusing on the reviewer's first point, we downloaded and processed 16 additional shRNA knockdown experiments followed by RNA-sequencing corresponding to *UPF1* and *UPF2*, which are known to be involved in NMD function (see Methods section: "*Analysing shRNA knockdown of RBPs followed by RNA-sequencing data from ENCODE*", Results section "*Accuracy in splicing is affected by RNA-binding protein (RBP) expression changes*", and Fig.5c). Results from this analysis indicated that mis-splicing activity was higher in knockdown experiments of both genes at both the acceptor and donor splice sites of the common introns studied. This increase in mis-splicing appeared to be relatively modest in comparison to those observed when spliceosome components or splicing regulators were knocked down, despite the essential role of *UPF1* and *UPF2* in degrading aberrant transcripts (see Fig.5c). We have incorporated this analysis in the revised manuscript and discuss the role of the NMD machinery in regulating splicing noise though perhaps less than might have been expected.

We also considered the analyses identified by the reviewer within the manuscripts Karousis et al. (2021) and Kovalak et al. (2021). To implement an analysis similar to that of Fig.5 from Karousis et al. (2021), we used shRNA knockdown data followed by RNA-sequencing from the ENCODE project and corresponding to 8 experiments against no target gene, namely control experiments, and 8 experiments against the two core NMD components *UPF1* and *UPF2*. Since we were working with intronic data in the context of this manuscript, we were not able to identify exons affected/not affected by the NMD action. Instead, we selected the annotated introns from protein-coding transcripts that were accurately spliced across the 8 control experiments but that had evidence of mis-splicing activity in any of the 8 knockdown samples targeting *UPF1* and *UPF2*. We classified these annotated introns as “NMD sensitive”. Likewise, we selected the annotated introns from control experiments that remained accurately spliced under knockdown conditions, and considered them as “NMD resistant”. We obtained an equal number of “NMD resistant” and “NMD sensitive” introns (N=4,184) by subsampling them on the basis of mean expression level similarity (Supplementary Figure 14a). We then compared mean conservation scores and motif sequence strength surrounding their two 5’ and 3’ splice sites. Mean conservation scores were calculated using phastCons17 (Siepel, A. et al. 2005) values by obtaining the mean phastCons17 scores overlapping the 100-bp intronic window sequence downstream and upstream of both splice sites. PhastCons17 represented conservation scores across 16 primate genomes and humans. Splice site strength was measured through MaxEntScan scores (Yeo and Burge 2004). Results of this analysis indicated that the annotated introns classified as “NMD sensitive”, were less conserved and weaker at their 3’ss compared to “NMD resistant” introns (phastCons17: paired one-tailed Wilcoxon Rank-sum test, effect-size=0.07, $P<0.001$; 3’ss MES: paired one-tailed Wilcoxon Rank-sum test, effect-size=0.03, $P=0.01$)(Supplementary Figure 14b,c). These results suggest that the NMD machinery has a role in degrading transcripts generated through mis-splicing of weak and poorly conserved acceptor splice sites.

2. Along the same lines, an essential conclusion of the study is the correlation between ageing and the availability of splicing-related factors. In this view, it is recommended to address whether mis-splicing is related to the abundance of NMD factors at different ages, as this would mean that the identification of mis-spliced isoforms is not attributed only to splicing deficiency but also to the reduced availability of NMD factors.

We thank the reviewer for this very useful comment, which we have implemented. We evaluated whether the expression levels of different well-known components of the NMD machinery (*UPF1*, *UPF2*, *SMG1*, *SMG6* and *SMG7*) were associated with age across the samples from different GTEx tissues. With this in mind, we built a linear regression model per NMD gene to predict their log₁₀ TPM levels across the samples of each tissue studied and used the age of the individual as one of the covariates included in the analysis (see Methods section “Changes in RBP expression levels with age”). Results from this analysis indicated that “age” was associated with reduced expression of components of the NMD factors studied across multiple tissues. This suggests that the reviewer is correct that at least a subset of the mis-splicing events detected with age could be due to a failure of the NMD machinery to adequately degrade mis-spliced transcripts. We have incorporated this analysis in the amended manuscript (see Methods section “Changes in RBP expression levels with age”, Results section “Increasing age is associated with increasing levels of mis-splicing”, Discussion section and Supplementary Figure 17)

3. The authors applied stringent filtering of reads to avoid including false-positive reads of splicing noise as described in the extended data Fig. 1. I would suggest including a short

description of these criteria in the first paragraph of the results where authors refer to the extended data fig.1, as they contribute to the robustness of the study.

We thank the reviewer for this comment. We have amended the manuscript and included a short description of the filtering criteria applied to the set of split reads used in this project. Changes correspond to the first paragraph of the Results section *“Novel donor and acceptor junctions are commonly detected and exceed the number of unique annotated introns by an average of 11-fold”*.

4. In an effort to implicate NMD, the authors present in Fig.5 factors that are erroneously stated as NMD factors (ribosomal proteins and general translation factors). This panel should be removed as these factors generally impact the transcriptome mostly in an NMD-independent manner.

We thank the reviewer for this comment. We have amended Fig.5 in the manuscript to replace the splicing factors previously stated as NMD factors with the knockdown experiments from *UPF1* and *UPF2* genes. We have only included these two NMD factors because the ENCODE platform only had shRNA knockdown RNA-sequencing experiments for these two genes with known NMD functions.

5. There are reports of short-distance tandem sites giving rise to coding proteins that contribute to the protein evolution that the authors may consider discussing. In such cases, one should be cautious before characterizing such events as mis-splicing. Some examples are the following: <https://doi.org/10.1371/journal.pbio.1001229>, <https://doi.org/10.1371/journal.pone.0096557>, <https://doi.org/10.1371/journal.pcbi.1008329>. Consider adding this notion to the discussion.

We thank the reviewer for this comment. We have amended the manuscript accordingly and added this notion to the *Discussion*. We have also referenced the studies provided.

6. On p.6 (last two lines), the sentence "High sequence similarity between novel and annotated splice sites might be expected if these sites were near each other" was a bit unclear at first read. I suggest replacing the word "near" with "in close proximity".

We thank the reviewer for this comment. We have amended the manuscript accordingly and replaced the "near" with "in close proximity" (Results section *“Novel junctions associated with protein-coding transcripts are predicted to be deleterious in 63.5% of cases”*).

7. Consider elaborating further on gene-level properties associated with mis-splicing (2nd paragraph of p.8), based on Fig.5.

We thank the reviewer for this suggestion, which we have applied. We have amended the manuscript and changes can be located in the Results section *“High sequence fidelity in the vicinity of exon-intron junctions is required to maintain splicing accuracy”*.

8. On Fig.4a the y axis description should be positioned lower since it refers to both panels.

We thank the reviewer for this comment. We have amended Fig.4a to position the y-axis label lower as suggested.

Reviewers' Comments to the Author

Re: Resubmission of Manuscript (NCOMMS-23-15679B, Submission #2)

We thank the three reviewers for their helpful reviews. We have considered all suggestions (see below) with the original comments in black and our detailed responses highlighted in blue.

Referee reports

Reviewer #1 (Remarks to the Author):

The authors have effectively addressed the primary technical and methodological points I raised, resulting in a significantly improved manuscript. While I have no reservations regarding the data quality, I maintain my view that the study's contribution to advancing the field is limited, and it lacks compelling evidence for biological relevance.

We thank the reviewer for the time and effort dedicated to reviewing our amended manuscript and for his/her recognition of the technical and methodological improvements in the quality of the work presented. However, we respectfully disagree with the reviewer concerning his/her opinion on the limited biological relevance and contribution of this study to the field more broadly.

To date, the studies addressing the role of RNA splicing and its dysregulation with age and in the context of Alzheimer's disease have been primarily focused on evaluating splicing defects occurring either in target genes due to specific splicing-altering variants, or by identifying genes which undergo large effects on transcript use (Balliu, B, et al. 2019; Baralle and Romano, 2023; Towfique et al. 2018; Dunhui Li, et al. 2021).

Since these approaches will miss small, but global effects on splicing accuracy, our study took a different approach. To understand transcriptome-wide changes in splicing fidelity and its drivers, we took the novel approach of prioritising introns and their properties within the context of i) changes in the abundance and activity of key RBP and NMD factors, ii) across ageing and iii) Alzheimer's disease (AD).

Our study has provided three major findings which, to the best of our knowledge, are not established. Firstly, we found that splicing errors occur in a highly consistent manner which reflects splicing mechanisms and the activity of splicing regulators. Secondly, we found that the human brain appears to be particularly sensitive to changes in the expression levels and activity of RBPs that regulate splicing. Thirdly, we noted that splicing factors play a particularly important role in global effects on splicing with age and AD. These findings have implications for our understanding of human brain diseases and will be of particular relevance for the development of splice-switching RNA-based therapies. To the best of our

knowledge, no existing studies address splicing accuracy from this point of view, nor have they drawn these results.

The authors emphasize the novelty of the concept of "splicing accuracy." However, it remains ambiguous how this concept differs from "mis-splicing," a phenomenon extensively documented in association with aging (see PMID: 36040386) and neurodegenerative diseases such as Alzheimer's disease (see PMID: 37336982).

We thank the reviewer for this comment. We agree with the reviewer regarding the potential ambiguity in the use of terms such as "splicing accuracy" and "mis-splicing".

We use the term "splicing accuracy" to refer to the study of splicing fidelity, primarily represented by unregulated errors that globally occur at low frequencies. On the other hand, "mis-splicing" has largely been used to refer to large effects on transcript use at specific genes associated with a disease state, and which are likely to be regulated events.

We have amended the text and added clarification to distinguish between these two terms. Changes can be found throughout the manuscript, but mostly in the Introduction and the Results section (*"Novel donor and acceptor junctions are commonly detected and exceed the number of unique annotated introns by an average of 11-fold"*).

Reviewer #2 (Remarks to the Author):

I appreciate the authors' efforts to address and improve the manuscript based on the comments. They carefully and sincerely replied to the comments along with extensive additional analyses.

We thank the reviewer for his/her kind comments and for dedicating the time to review our amended manuscript. We also thank the reviewer for recognising our efforts in addressing all comments resulting in an improved manuscript.

However, the increase of mis-spliced transcripts over aging and diseases is significantly associated with the abundance of NMD factors, which are already documented multiple times before. It is still unclear how much or which splicing factors (RBPs) contribute the mis-splicing over the conditions. Without controlling NMD factor changes or controlled experimental validation, it is difficult to argue how much or which splicing factors (RBPs) impact the mis-splicing across age and disease.

Focusing on the concerns regarding the possible contribution of the NMD machinery to age-related changes in splicing accuracy, we have run additional analyses to calculate the impact of age on the MSR values (our measure of splicing inaccuracy) while considering expression levels of the NMD machinery across age groups. We extended this approach to study each of the three RBP classes outlined in our manuscript (i.e. "Splicing Regulation", "Spliceosome", and "Exon-Junction Complex"). This allowed us to assess the relative impact of NMD as compared to RBP splicing factors.

To do this, we focused on three GTEx tissues, blood, blood vessel and brain, based on their high age-effect sizes as reported in the original version of this manuscript (Fig7.a). We calculated the fold-change in TPM expression of each RBP/NMD factor in the '60-79' year category as compared to the '20-39' yr category per GTEx tissue and corrected the MSR values accordingly. As previously noted in the original manuscript, both RBP and NMD factors tended to decrease in expression with increasing age. We then repeated the assessment of age differences in MSR values between the two age groups using the Wilcoxon Effect Size method. Finally, to assess the impact of each RBP/NMD factor on MSR values, we ranked the factors in terms of their contribution to age-related splicing fidelity (i.e. age-related effect size).

Results from this analysis indicated two main findings. First, we noted that after the correction of MSR values, age effect sizes tended to decrease as compared to those before correction, and this was most evident in brain tissue. This finding indicates that correcting for RBP expression changes with age decreases the apparent effect of age on splicing noise at both splice sites, suggesting that indeed these factors contribute to splicing inaccuracy

with age (**Supplementary Figures 19-21**). Second, when we ranked individual RBPs and NMD factors in terms of the extent to which correcting for their expression reduced the effect of age on splicing inaccuracies, we noted the importance of RBPs. More specifically, we found that the top-ranked genes (top 5) for all three tissues were RBPs involved in splicing (**Supplementary Table 8-10**).

We extended this type of analysis to study Alzheimer's disease. As described above, we evaluated the relative contributions of the NMD machinery and RBPs to splicing inaccuracies in the disease state. The results from this analysis indicated a decrease in splicing noise with disease after correction for the relative expression of each RBP/NMD factor (**Supplementary Figure 22**), with RBPs again being top-ranked (**Supplementary Table 13**).

Thus, overall, these analyses suggested that while NMD likely contributes to the presence of inaccurately spliced transcripts with increasing age and in AD, the expression of RNA splicing factors is likely to be more important.

Reviewer #3 (Remarks to the Author):

After an intense peer review of an already interesting manuscript, the revised manuscript of García-Ruíz et al. has been considerably improved, and I support its publication in Nature Communications.

We thank the reviewer for appreciating the value of the manuscript and for recommending this manuscript for publication in Nature Communications.

Addressing comments 1 and 2, the authors performed additional analysis and reported important evidence concerning the relationship between mis-splicing and NMD. Even though it is not the main focus of the manuscript, this is valuable evidence related to the broad community of genetic diseases. Therefore, I suggest highlighting their findings in the discussion and, if possible, illustrating this aspect in the Abstract and the Introduction. More specifically, I suggest removing the phrase from the discussion and mentioning that authors do not consider NMD in their analysis (Fifthly, our analyses have not modelled the direct impact of NMD on mis-splicing activity). Instead, in light of their new analysis, I suggest pointing out that knocking down components of the NMD pathway led to a modest increase in the levels of acceptor splice-site noise and that ageing may be related to variation in NMD factors expression levels. Previous evidence pointing in a similar direction is the following: <https://doi.org/10.1186/s13059-021-02309-y>, <https://doi.org/10.1186/s13059-021-02439-3>, <https://doi.org/10.1371/journal.pgen.0020045>. I would like to commend the interesting work and review of the authors.

We thank the reviewer for his/her kind words and for recognising the value of this manuscript more broadly. We have followed the reviewer's recommendations and amended the text accordingly.

Focusing on the reviewer's first suggestion, we have followed his/her recommendations and added to the Discussion the notion that knocking down components of the NMD pathway led to a significant but modest increase in acceptor splice-site noise.

Focusing on the reviewer's second suggestion, we have run additional analyses to understand the potential relationship between increasing ageing, variation in the expression of NMD factors and changes in splicing noise. More specifically, we evaluated the relative contribution of individual NMD and RBP factors to inaccurate splicing with increasing age in blood, blood vessel and brain (**Supplementary Figure 19-21**). When we ranked factors in terms of their contribution to age-related splicing fidelity, we noted that the top-ranked genes (top 5) for all three tissues, were RBPs involved in splicing (**Supplementary Table**

8-10). We then repeated this analysis in the disease state (AD vs unaffected individuals, dataset GEO:GSE95587). Again, we found a decrease in splicing noise with disease after correction for the relative expression of each RBP/NMD factor (**Supplementary Figure 22**), with RBPs again being top-ranked (**Supplementary Table 13**).

These analyses suggested that while NMD likely contributes to the presence of inaccurately spliced transcripts with increasing age and in Alzheimer's disease, the expression of RNA splicing factors is likely to produce a more profound effect. We have explained this in both the Introduction and Discussion sections.

Reviewers' Comments to the Author

Re: Resubmission of Manuscript (NCOMMS-23-15679B, Submission #3)

We thank the reviewer for their helpful comments. We have considered their suggestions (see below) with the original comments in black and our detailed responses highlighted in blue. We also note the editor's comments advising that the remaining concerns should be addressed in the form of textual clarifications with the addition of a discussion of the limitations and need for future studies.

Referee reports

Reviewer #2 (Remarks to the Author):

Thank you for the repeated explanation and additional analysis for claiming that splicing-related RBPs give more impact on splicing inaccuracy over aging while NMD factors also contribute somewhat. However, I am still concerned about which RBPs are actually involved in the splicing accuracy of which genes. If all decreased RBPs contribute to splicing inaccuracy, then how can the mis-splicing of a particular gene be targeted to treat disease?

On the other hand, the authors' claim may be strengthened if they can analyze how much the non-NMD targets are affected by aging or by the decrease of splicing-related RBPs. To collect the non-NMD targets, they can consider publicly available RNA-seq data followed by UPF1 KD or KO.

Focusing on the Reviewer's first concern, namely our claim that changes in the expression of splicing-related and spliceosomal RBPs produce a greater impact on splicing accuracy than the NMD factors, we have amended our manuscript to avoid any overstatement (text changes can be found in the Introduction, Results and Discussion sections as highlighted in the new submission). In addition, we have discussed the need for further research to elucidate the specific impact of each RBP and NMD factor on the splicing accuracy of particular genes.

We also appreciate the reviewer's suggestion of using *UPF1* knockdown RNA-seq data to strengthen our study. In fact, we have already analysed RNA-sequencing data from both *UPF1* and *UPF2* knockdown experiments in cell lines and assessed inaccurately spliced introns that appeared to be resistant to NMD in terms of their sequence conservation and splice site motif strength (**Supplementary Figure 14**). We also used knockdown data from *UPF1*, *UPF2*, as well as from other 56 RBPs to compare the levels of splicing inaccuracies in annotated introns in knockdown experiments as compared to their isogenic controls (**Fig. 5c**). While these results overall suggested the greater importance of RBPs in regulating transcript integrity, we recognise that additional experiments and analyses are required to determine which specific RBPs are needed to ensure accurate splicing of specific introns. We have now made this clear in the amended discussion.